# Knowing Who, Not How Much: Learning-Augmented Mechanisms for Consumer Utility Maximization

Kira Goldner [1]  Divyarthi Mohan [2]  Thodoris Tsilivis [1]

## Abstract

We study consumer utility maximization in an online random-order model where strategic agents arrive sequentially. To circumvent strong impossibility results for utility maximization, we turn to the framework of learning-augmented mechanism design. Crucially, we show that the types of predictions commonly used in learning-augmented mechanism design (such as predictions of agent values or the optimal value) are not useful for utility maximization, where payments are directly at odds with the objective. Instead, we identify that a qualitatively different kind of prediction suffices: the identity of the highest-valued agent. First, we provide a deterministic truthful mechanism for our online setting by adapting offline randomized techniques. Then, we augment our mechanism with predictions. When the predictions are correct, we achieve a constant approximation to the optimal solution under full information (consistency), and even when predictions are arbitrarily bad, we guarantee a constant approximation to the best implementable solution (robustness).

## 1. Introduction

Consider a simple resource allocation problem, where a social planner must allocate a single item to one of $n$ agents with private values $v_1 \geq v_2 \geq \cdots \geq v_n$ for being allocated. If the values were publicly known, the welfare-maximizing allocation is trivial: allocate the item to the highest-valued agent, obtaining utility $v_1$. However, the social planner cannot distinguish the agents without incentivizing them to report their values through the use of some sort of payments. For instance, the classical second-price auction incentivizes truthful reporting by allocating to the highest bidder and charging them the second highest bid (Vickrey, 1961). Yet

in many settings, charging monetary payments is infeasible or undesirable. We instead use other forms of payments or "ordeals," such as wait times, bureaucracy like filling out paperwork, or reductions in service (such as in cloud computing). These non-monetary payments are not transferable, but rather are "burnt," and exist only as a tool to elicit information. Thus, in these settings, the social planner's goal is actually to maximize the consumer utility (or "residual surplus"): the value obtained for the allocation minus the cost incurred by the agents.

The objective of consumer utility maximization highlights the tension between allocative efficiency and the cost of elicitation. While the second-price auction obtains optimal welfare $v_1$, the consumer utility is only $v_1 - v_2$, as the winner suffers a cost in order to be distinguished from the rest of the agents. When $v_1$ and $v_2$ are close, nearly all of the surplus is consumed by the elicitation cost. On the other hand, a simple lottery that allocates the item uniformly at random without imposing any payments cannot distinguish the high-valued agent from the rest. The expected consumer utility $\sum_i v_i/n$ can be bad when only a few agents have high value. This tension between elicitation and efficiency leads to a strong impossibility result for utility maximization. No truthful mechanism, even with full distributional knowledge, can guarantee better than an $O(\log n)$-approximation to the optimal social welfare (i.e., the first-best benchmark with complete information) (Hartline & Roughgarden, 2008). The objective of utility maximization has been under-explored by the algorithmic community, but even in these limited works, this logarithmic barrier repeatedly crops up (Appendix A).

**Mechanisms with Predictions.** In order to circumvent this impossibility result, we turn to the framework of mechanism design with predictions (e.g., (Xu & Lu, 2022; Agrawal et al., 2022)). Suppose the social planner now has access to predictions about the values of the agents, e.g., from an ML model or other black-box source. Can we avoid exorbitant elicitation costs while still obtaining good allocative efficiency? Crucially, as with LLMs, these predictions could be perfectly accurate, or they might be unreliable hallucinations. Hence, we seek to design mechanisms with best-of-both-worlds guarantees: (1) consistency, or achieving

[1]Boston University [2]Columbia University . Correspondence to: Thodoris Tsilivis <tsilivis@bu.edu>.

*Proceedings of the 43rd International Conference on Machine Learning*, Seoul, South Korea. PMLR 306, 2026. Copyright 2026 by the author(s).

constant-factor approximation by harnessing correct predictions, and (2) robustness, or matching worst-case guarantees when predictions are imperfect.

A natural first question that arises is: what predictions suffice, or how much information granularity is needed to achieve such approximation guarantees? Prior works on mechanism design with predictions have studied other objectives (e.g., revenue and welfare) and considered a variety of prediction models, from complete value predictions about each agent (Xu & Lu, 2022; Caragiannis & Kalantzis, 2024; Lu et al., 2024) to predictions about the optimal solution, i.e., highest value (Antoniadis et al., 2023; Agrawal et al., 2022; Balkanski et al., 2024). Unfortunately, the latter does not help for our objective of utility maximization, as Qiao & Valiant (2023) showed that it is not possible to obtain better than a $\Omega(\log n)$-approximation even with *perfect predictions* about the highest value.[1] This is because the mechanism still cannot identify which agent holds this value without resorting to payments, which are directly at odds with our objective. Hence in this paper, we perform the first dedicated analysis on predictions-augmented mechanism design for utility maximization.

See Appendix A for expanded details on these and other related works.

**Our Work.** The goal of this paper is to design mechanisms that maximize utility when agents arrive and decisions must be made online. We restrict our attention to designing deterministic mechanisms, as in practice, consumers find randomization unpalatable (Liu et al., 2025; Tversky & Kahneman, 1992). Finally, we seek to explore the power of prediction in online deterministic utility-maximizing mechanism design. How much utility do (potentially incorrect) predictions enable us to capture? Can we give best-of-both-worlds guarantees, giving guarantees for when the predictions are both correct and incorrect? What benchmarks should we use? What sort of predictions are best-suited to this task? How does it differ from offline settings or settings with other objectives?

We answer all of these questions. In Section 3, we work toward our eventual robustness guarantee by designing a deterministic mechanism that guarantees a constant-factor approximation to $\mathrm{OPT}_L$ (our robustness benchmark). We do so by converting an offline randomized mechanism into an online deterministic mechanism and proving that the guarantees continue to hold.

In Section 4, we first explore the different possible predictions and show that a qualitatively different type of predic-

tion suffices: knowing who has the highest value, rather than what that value is. We then augment our deterministic mechanism with predictions, and compare methods to show that there is one simple way to do so that achieves a best-of-both-worlds guarantee: both consistency (constant-approximation to the first-best benchmark when predictions are correct) and robustness (constant-approximation to the best implementable solution).

## 2. Preliminaries

We consider the setting of auctioning off a single item to $n$ agents. Each agent $i$ has a private value $v_i$ for the item. We study the prior-free (or worst-case) setting, in which the mechanism has no prior information about the values $v_i$; they may be any input instance. We assume, without loss of generality, that the agents are indexed from highest to lowest value, that is, $v_1 > \cdots > v_n$, where index $i$ denotes the agent with the $i$-th highest value, and that these values are distinct.

The $n$ agents arrive online according to a uniformly-at-random permutation $\sigma$—this is the random-order model for online arrival. As the agents arrive, they report their bid to the mechanism, and an irrevocable online decision regarding allocation and payment is made. We use $\sigma(i)$ to denote the $i$-th agent that arrives according to the permutation. We let $[a : b]$ denote the set integers between (and including) $a$ and $b$. Then $\sigma[1 : n]$ denotes the entire arrival sequence of the agents.

An agent $i$ who is allocated an item at price $p$ obtains utility $v_i - p$, where $v_i$ is their value for the item. We study dominant-strategy incentive compatible mechanisms, in which for any agent $i$, it is a utility maximizing strategy for them to bid their true value $v_i$, regardless of what all other agents report. Additionally, we require individual rationality, which imposes the utility of all agents to be non-negative.

Our objective is consumer utility maximization. In the single-item setting, this is the (expected) utility of the agent who is allocated the item. We consider two benchmarks to compare our mechanisms' utility guarantees against:

1. **First-Best** is the maximum achievable welfare under full information, i.e., if the mechanism observes agents' true values and agents do not misreport. Formally, $\mathrm{FB} = v_1$.

2. **Optimal-lottery** is the highest expected utility that can be achieved by a mechanism that posts a price $p$ to the agents, randomly allocates to any one of them that reported a value higher than $p$, and charges that agent

---

[1]In fact, achieving a constant approximation to the first-best remains impossible even when the correct predictions are about the complete multi-set of all values (without knowing which agent has what value) for the random order arrival model.

a price of $p$. Formally:

$$\text{OPT}_L = \max_{p \in \mathbb{R}_+} \frac{1}{|\{i \,:\, v_i > p\}|} \sum_{i:v_i>p} (v_i - p).$$

We note that $\text{OPT}_L$ is a natural benchmark for the prior-free setting, and for a single-item auction, is in fact exactly the same as the prior independent benchmark $\mathcal{G}(\vec{v})$ used in (Hartline & Roughgarden, 2008). This is because $\text{OPT}_L$ is the best utility achievable by any implementable mechanism. Moreover, the gap between FB and $\text{OPT}_L$ (and thus any implementable mechanism) is $\Omega(\log n)$ in the worst case.

Our main research agenda revolves around answering the question: what types of predictions can surpass tight performance barriers in single-item utility-maximizing online auctions?

In the learning-augmented mechanism design literature, performance is measured in terms of *robustness* and *consistency*. For our work, we define these as follows:

- $\gamma$-**Consistency:** For $\gamma \in [0,1]$ a mechanism is $\gamma$-consistent if it obtains an $\gamma$-approximation ratio to FB when the predictions are correct.

- $\rho$-**Robustness:** For $\rho \in [0,1]$, a mechanism is $\rho$-robust if it obtains an $\rho$-approximation ratio to $\text{OPT}_L$ under arbitrarily erroneous predictions.

## 3. Competitive Mechanisms for Online Random Order Arrival

In this section, we present a deterministic online mechanism that obtains a constant approximation to the optimal lottery $\text{OPT}_L$ (our robustness benchmark) under the random-order model. In Section 4, we will subsequently adapt this mechanism in order to design a learning-augmented mechanism that leverages predictions and additionally achieves a constant consistency guarantee to the first-best benchmark FB.

### 3.1. Warm-up: A Randomized Online Mechanism

In this subsection, we present a *randomized* mechanism for the single-item utility maximization problem, assuming random arrival order of the agents. We use the offline Random Sampling Optimal Lottery (RSOL) mechanism by Hartline & Roughgarden (2008) as our starting point. RSOL is a truthful mechanism that obtains a constant approximation to $\text{OPT}_L$, the optimal $p$-lottery. We propose a modification[2] of RSOL for the online random-order model, instead of

---

[2]The original RSOL mechanism runs Vickrey on $S$ with the same probability that we offer $p_{\max}(\bar{S})$ to $S$. When $v_2 \in \bar{S}$, this is equivalent, so their analysis holds identically for this modification.

drawing on the online algorithm for pen testing in (Qiao & Valiant, 2023). This is for two reasons: (1) we want a meaningful constant-approximation to $\text{OPT}_L$ (rather than a $\log n$-approximation to FB), and (2) we believe RSOL is a simpler mechanism that is well-suited for derandomization.

**Modified RSOL (Hartline & Roughgarden, 2008).** The agents are randomly partitioned into two sets—the sample set $\bar{S}$, and the execution set $S$—where each agent is placed independently into either set with probability $\frac{1}{2}$. The mechanism then computes two candidate lottery prices:

- $p_{\max}(\bar{S})$: The maximum value in the sample $\bar{S}$. That is, $p_{\max}(\bar{S}) := \max_{i \in \bar{S}} v_i$.

- $p_{\text{lot}}(\bar{S})$: The lottery price that maximizes utility among the agents in the sample $\bar{S}$. That is,

$$p_{\text{lot}}(\bar{S}) = \underset{p \geq 0}{\operatorname{argmax}} \frac{\sum_{i \in \bar{S}} (v_i - p)^+}{|\{i \in \bar{S} : v_i > p\}|}.$$

The mechanism then tosses a fair coin to set $p(\bar{S})$[3] as one of these two prices, and executes a $p(\bar{S})$-lottery on the execution set $S$.

**Coin-toss Mechanism $\mathcal{M}_{\text{toss}}$.** For a uniformly random arrival order $\sigma$, we draw $t$ from $\text{Bin}(n, \frac{1}{2})$, and simply observe the first $t$ arriving agents as a sample $\bar{S} \leftarrow \sigma[1:t]$; we do not serve them. Then, we toss a fair coin to set $p(\bar{S})$ as either $p_{\max}(\bar{S})$ or $p_{\text{lot}}(\bar{S})$ (as in modified RSOL). We then offer the item to any agent after $t$ for the posted price $p = p(\bar{S})$ and allocate to the first agent who accepts the price.

We argue that the coin-toss mechanism $\mathcal{M}_{\text{toss}}$ achieves the same guarantees as Modified RSOL when the agents arrive online in a uniformly random order $\sigma$ by noting the following.

First, the induced distribution over the sample set $\bar{S}$ in mechanism $\mathcal{M}_{\text{toss}}$ is the same as the distribution of the sample set in RSOL.

Second, both mechanisms use a fair coin to decide which price to post on the execution set. Thus, the distribution of the price $p$ is exactly the same.

Finally, we crucially observe and leverage the equivalence between random order posted price mechanisms and offline $p$-lotteries. An offline $p$-lottery uniformly-at-random selects some agent in $S$ that reported a bid larger than the price $p$ to allocate to and charge $p$. Equivalently, an online posted price mechanism with price $p$ accepts the first (random) agent in $S$ that reports a bid larger than the price $p$, allocating and charging $p$. Due to this equivalence, we sometimes conflate random-order $p$ posted prices with $p$-lotteries.

---

[3]We will drop $\bar{S}$ for brevity when clear from context.

As a result, we have argued the same distribution over outcomes for both mechanisms, meaning that the analysis and guarantees of (modified) RSOL extend to the coin-toss mechanism $\mathcal{M}_{\text{toss}}$. For completeness, we state (and refer to (Hartline & Roughgarden, 2008)) the guarantees of $\mathcal{M}_{\text{toss}}$ (equivalently RSOL).

**Theorem 1.** *The randomized mechanism $\mathcal{M}_{\text{toss}}$ for the random-order setting achieves a constant $\frac{1}{1250}$-approximation to $\text{OPT}_L$, the expected utility of the optimal p-lottery.*

### 3.2. A Deterministic Online Mechanism

We start by discussing how randomness is used by the coin-toss mechanism $\mathcal{M}_{\text{toss}}$ and the adjustments we make to derandomize it and create the deterministic mechanism $\mathcal{M}_{\text{phase}}$.

The randomized mechanism makes exactly two random choices.

The first choice concerns where to split the arriving sequence between sample and execution. In $\mathcal{M}_{\text{toss}}$, the partition point $t$ is drawn from a binomial distribution so that the set of agents observed before posting a price has size $\frac{n}{2}$ *in expectation*. In $\mathcal{M}_{\text{phase}}$, we instead fix this split deterministically at $\frac{n}{2}$.

The second choice concerns which price to post. In $\mathcal{M}_{\text{toss}}$, a fair coin determines whether the mechanism posts $p_{\max}$ or $p_{\text{lot}}$. In $\mathcal{M}_{\text{phase}}$, we instead adapt this by posting both prices at different times, which leads naturally to a three-phase structure: a sample phase, followed by a max phase in which the price is $p_{\max}$ (the max value observed in the initial sample phase), and finally a lottery phase in which the price is $p_{\text{lot}}$ (the optimal lottery price over the observed values in both the first two phases). More specifically, the first half of the arrival sequence is itself split in secretary style between the sample and max phases, while the lottery phase occupies the remaining half.

These modifications create qualitative differences between the randomized mechanism $\mathcal{M}_{\text{toss}}$ and the deterministic mechanism $\mathcal{M}_{\text{phase}}$. First, the $p_{\max}$ and $p_{\text{lot}}$ phases are no longer each executed with probability $\frac{1}{2}$. Additionally, the sets on which these prices are evaluated no longer coincide (as random variables) with the execution set $\bar{S}$ in $\mathcal{M}_{\text{toss}}$. Nevertheless, the overall proof strategy remains close in spirit to that of (Hartline & Roughgarden, 2008); the main additional work is to show that the structural properties needed for the lottery-phase analysis continue to hold under this specific partition of the arrival sequence.

### 3.2.1. THE DETERMINISTIC MECHANISM

We now formally define the deterministic mechanism $\mathcal{M}_{\text{phase}}$ and proceed with the main theorem.

**Deterministic Phases Mechanism $\mathcal{M}_{\text{phase}}$.** Given a sequence of $n$ agents arriving in random order permutation $\sigma$ $(v_1, v_2, \ldots, v_n)$, the mechanism progresses in three phases:

1. **Sample phase** $N_{\text{sample}} \leftarrow \sigma[1 : \frac{n}{2e}]$. Observe but do not serve the first $n/2e$ agents, using them to compute $p_{\max} \leftarrow \max_{i \in N_{\text{sample}}} v_i$.

2. **Max phase** $N_{\max} \leftarrow \sigma[\frac{n}{2e} + 1 : \frac{n}{2}]$. Post a price of $p_{\max}$ to these agents and allocate to the first agent who accepts (if any). If no agent accepts, denote $\overline{N_{\text{lot}}} = \sigma[1 : \frac{n}{2}]$, and compute $p_{\text{lot}} \leftarrow p_{\text{lot}}(\overline{N_{\text{lot}}})$ the optimal p-lottery for agents in $\overline{N_{\text{lot}}}$ (those in the first two phases).

3. **Lottery phase** $N_{\text{lot}} \leftarrow \sigma[\frac{n}{2} + 1 : n]$. For the remaining agents in $N_{\text{lot}}$, post price $p_{\text{lot}}$ and allocate to the first bidder who accepts (with value $v_i > p_{\text{lot}}$).

**Theorem 2.** *The deterministic mechanism $\mathcal{M}_{\text{phase}}$ for the random-order setting achieves a constant $\frac{1}{625e}$-approximation to $\text{OPT}_L$, the expected utility of the optimal p-lottery.*

Before the theorem's proof, we lay out the underlying idea originating from (Hartline & Roughgarden, 2008), emphasizing where our analysis inevitably diverges. The key point is that the optimal p-lottery may derive its utility either from a large gap between $v_1$ and $v_2$, in which case it allocates to $v_1$ at a price of $v_2$ (captured by the max phase), or from a lower lottery price that extracts utility from a broader pool of agents (captured by the lottery phase).

The main technical challenge lies in adapting the lottery-phase analysis. As with RSOL, the argument relies on a balanced property between the first half of the permutation ($\overline{N_{\text{lot}}}$) and the lottery phase ($N_{\text{lot}}$). When the balanced property holds, both sets are representative of the full instance, implying that any posted price yields in expectation approximately the same utility whether evaluated on $\overline{N_{\text{lot}}}$, on $N_{\text{lot}}$, or on the full instance. Using a probabilistic coupling between the random-subset process in RSOL and our random-permutation process, we show that this property holds in $\mathcal{M}_{\text{phase}}$ with probability at least as large as in $\mathcal{M}_{\text{toss}}$ (Lemma 2). We now formalize with definitions and proceed to the proof of the corresponding balancedness guarantee.

Let $n_i = |N_{\text{lot}} \cap \{1, 2, \ldots, i\}|$ (and $\bar{n}_i = |\overline{N_{\text{lot}}} \cap \{1, 2, \ldots, i\}|$) be the number of agents that were placed in $N_{\text{lot}}$ (and $\overline{N_{\text{lot}}}$ respectively) out of the top $i$ valued agents. The *random order balanced property* that we will use is defined as event $E_{RO} = \{\forall i \geq 2 : \frac{1}{6} i \leq n_i \leq \frac{5}{6} i\}$, which

corresponds to the permutations with the property that for any $i \geq 2$, the fraction of the top $i$ agents that landed in $N_{\text{lot}}$ is bounded in $[\frac{1}{6}, \frac{5}{6}]$. Note that we could have equivalently expressed this event with $\bar{n}_i$, as by symmetry, whenever the property holds for $N_{\text{lot}}$, it must also hold for $\overline{N_{\text{lot}}}$. Thus, one intuitive way to think about this event is that it guarantees that *both* sets $N_{\text{lot}}$ and $\overline{N_{\text{lot}}}$ have at least a constant fraction of the top $i$ agents (for any $i > 2$).

We now discuss RSOL's balanced property. First define $s_i = |S \cap \{1, 2, \ldots, i\}|$ similarly to $n_i$ (where $S$ is a uniformly random subset) and define RSOL's balanced property as event $E_{RS} = \{\forall i \geq 2 : \frac{1}{6}i \leq s_i \leq \frac{5}{6}i\}$, which corresponds to any subset $S$ with the property that for any $i \geq 2$, the fraction of the top $i$ agents that landed in the subset $S$ is bounded in $[\frac{1}{6}, \frac{5}{6}]$. From (Hartline & Roughgarden, 2008), we get the following lemma about the probability of the balanced property conditioning on agents 1 and 2 being in subset $S$ and subset $\bar{S}$ respectively:

**Lemma 1** (Adapted from (Hartline & Roughgarden, 2008))**.** *Let $S$ be a uniform random subset of $[n]$. Then:*

$$\Pr_S\left[E_{RS} \mid s_1 = 1, s_2 = 1\right] \geq \frac{4}{5}.$$

For the analysis of the mechanism $\mathcal{M}_{\text{phase}}$, we require a similar claim for the random partition that it employs. Specifically, when the sets are partitioned into the first and second halves of the random arrival, as opposed to uniformly at random, the probability of this event only increases.

**Lemma 2.** *For a uniformly random order permutation $\sigma$ and a random subset $S$ it holds that:*

$$\Pr_\sigma\left[E_{RO} \mid n_1 = 1, n_2 = 1\right] \geq \Pr_S\left[E_{RS} \mid s_1 = 1, s_2 = 1\right].$$

We will prove this lemma via a coupling argument that compares the two partitioning procedures through a family of intermediate random processes. To motivate the coupling, we first compare the random subset event $E_{RS}$ and the random permutation event $E_{RO}$ at a high level. Putting randomness aside for a moment, both events impose the same balancedness condition on a partition of the agents: for every $i$, the top-$i$ agents should be split so that neither side contains too small or too large a fraction (e.g., between $i/6$ and $5i/6$). Then, one way to reason about balancedness is to track the (absolute) difference between the sizes of the two sets as each element $i$ is placed in the partition. This induces a stochastic process that can be viewed as a random walk on the non-negative integers (initialized at 0), where a $+1$ step corresponds to placing the next element on the currently larger set, and a $-1$ step corresponds to placing it on the smaller set. Under this view, balancedness corresponds to the event that this walk never crosses a time-varying barrier (which we will later compute to be $2i/3$)

that captures precisely when the imbalance between the partitioning sets becomes too large.

To account for the randomness associated with the partitioning process, we simply need to specify the random walk transition probabilities for each partitioning process. In the random-subset process ($\mathcal{M}_{\text{toss}}$), each step of the walk is an independent move of $\pm 1$ with probability $1/2$. In contrast, in the random order process ($\mathcal{M}_{\text{phase}}$), the transition probabilities are history-dependent: at any point, the next assignment is biased toward the smaller set. Now notice that if the two processes are at the same state after some number of steps, the random-order walk is more likely to move toward 0 and less likely to move toward the imbalance barrier. This single-step comparison captures the key intuition, but is not sufficient on its own to compare the full processes because of the history-dependence. This is precisely why we introduce intermediate processes, allowing us to compare the two processes at one step at a time (while sequentially conditioning on history), ultimately showing that balancedness under random order is more likely than under random subsets.

The above may seem redundant given that we only wish to prove a lower bound for the random order process and the event $E_{RO}$. We emphasize, however, that reasoning about $E_{RO}$ directly is considerably more difficult. The main hurdle is that the balancedness condition must hold for the top $i$ agents for *all* $i \geq 2$—it is not a single concentration bound at one fixed $i$, and further, a clean induction proof is prevented by obvious dependencies between the condition for varying $i$. The full proof of Lemma 2 is deferred to Appendix B.1.

We are now ready to prove Theorem 2.

*Proof of Theorem 2.* To prove the approximation guarantee of the mechanism, we need to analyze the guarantees of each contributing phase of the mechanism: $N_{\text{max}}$ and $N_{\text{lot}}$. Note that for the lottery phase to contribute, it must be the case that the item was not sold during the max phase $N_{\text{max}}$, so we must condition on this in our analysis.

**Lottery phase analysis:** We begin by analyzing the guarantees of the *lottery* phase $N_{\text{lot}}$. We notice first that in any ordering $\sigma$ that has agent 1 in the third (lottery) phase $N_{\text{lot}}$ and agent 2 in the first (sample) phase $N_{\text{sample}}$, mechanism $\mathcal{M}_{\text{phase}}$ is *guaranteed* to reach the lottery phase $N_{\text{lot}}$—no one can afford a price of $p_{\text{max}} = v_2$ except agent 1. Hence, to bound the contribution of the lottery phase, we condition on this event along with the event that the balanced property holds $E_{RO}$, defining $E_{lot} = E_{RO} \cap \{1 \in N_{\text{lot}} \text{ and } 2 \in N_{\text{sample}}\}$. In the following lemma, we relate the expected utility of $\mathcal{M}_{\text{phase}}$ conditioned on $E_{lot}$ to $\text{OPT}_L$ (where expectation and probabilities are over the uniformly random permutation $\sigma$).

**Lemma 3.** *Conditioning on event $E_{lot}$, the expected utility of $\mathcal{M}_{\text{phase}}$ is at least $\frac{\text{OPT}_L - (v_1 - v_2)}{125}$.*

The proof parallels (Hartline & Roughgarden, 2008) and can be found in Appendix B.2. At a high level, the lemma follows because the balanced property $E_{RO}$ imposes that both sides of the partition are representative of the full set of agents in the following sense: any price posted in one part of the partition yields approximately the same guarantees when posted to the other part, as well as to the full instance (Lemma 6 in Appendix B.2). Thus, the optimal $p$-lottery of one side of the partition maintains its guarantees when posted on the other side, while also remaining competitive against the global optimum (Lemma 7 in Appendix B.2).

To wrap things up, we quantify the probability of this desirable event $E_{lot}$. We first lower-bound the auxiliary probability of $\Pr[1 \in N_{\text{lot}} \text{ and } 2 \in N_{\text{sample}}] = \frac{1}{2} \cdot \frac{\frac{n}{2e}}{n-1} \geq \frac{1}{4e}$. Finally, we lower-bound the probability of event $E_{lot}$:

$$
\begin{aligned}
\Pr[E_{lot}] &= \Pr[E_{RO} | 1 \in N_{\text{lot}} \text{ and } 2 \in N_{\text{sample}}] \\
&\quad \cdot \Pr[1 \in N_{\text{lot}} \text{ and } 2 \in N_{\text{sample}}] \\
&= \Pr[E_{RO} | n_1 = 1, n_2 = 1] \\
&\quad \cdot \Pr[1 \in N_{\text{lot}} \text{ and } 2 \in N_{\text{sample}}] \\
&\geq \frac{4}{5} \cdot \frac{1}{4e} = \frac{1}{5e},
\end{aligned} \tag{1}
$$

where the second equality follows because Lemma 2 holds for any ordering with $\{n_1 = 1, n_2 = 1\}$ which is true under $\{1 \in N_{\text{lot}} \text{ and } 2 \in N_{\text{sample}}\}$, and the inequality is due to Lemmas 1 and 2.

**Max phase analysis:** We now explain how the max phase contributes to the mechanism's expected utility. Let $E_{max}$ denote the event that the max phase of the mechanism allocates to agent 1, the highest valued agent, at any price. Notice that under this event, the max phase yields utility at least $v_1 - v_2$. We decompose $E_{max}$ into the intersection of two independent events: i) the event $\{1 \in \overline{N_{\text{lot}}}\}$, and ii) the event $E_{sec}$ that the max phase allocates to the highest valued agent in $\overline{N_{\text{lot}}}$ (this need not necessarily be agent 1). We compute the probability of $E_{max}$:

$$
\begin{aligned}
\Pr[E_{max}] &= \Pr[E_{sec} \cap 1 \in \overline{N_{\text{lot}}}] \\
&= \Pr[E_{sec}] \cdot \Pr[1 \in \overline{N_{\text{lot}}}] \\
&= \frac{1}{2e}.
\end{aligned} \tag{2}
$$

where the probability of $E_{sec}$ follows from the standard secretary problem analysis Dynkin (1963). Note that $E_{sec}$ only depends on the ordering of the agents in $\overline{N_{\text{lot}}}$, irrespective of the realization of $\overline{N_{\text{lot}}}$ (and is thus independent of $\{1 \in \overline{N_{\text{lot}}}\}$).

We are now ready to compute the expected utility of the mechanism:

$$
\begin{aligned}
U_{\mathcal{M}_{\text{phase}}} &= \mathbb{E}[\text{Utility of } \mathcal{M}_{\text{phase}}] \\
&\geq \mathbb{E}[\text{Utility of } \mathcal{M}_{\text{phase}} | E_{max}] \cdot \Pr[E_{max}] \\
&\quad + \mathbb{E}[\text{Utility of } \mathcal{M}_{\text{phase}} | E_{lot}] \cdot \Pr[E_{lot}] \\
&\geq (v_1 - v_2) \cdot \frac{1}{2e} + \frac{\text{OPT}_L - (v_1 - v_2)}{5} \cdot \frac{1}{25} \cdot \frac{1}{5e} \\
&= \frac{2\text{OPT}_L + 623(v_1 - v_2)}{1250e} \\
&\geq \frac{\text{OPT}_L}{625e},
\end{aligned} \tag{3}
$$

where for the first inequality we have used the law of total expectation and the non-negativity of the utility objective, and the second inequality follows from Equation (2) for the first term, and Equation (1) and Lemma 3 for the second term. $\square$

# 4. Prediction-Augmented Mechanisms for Utility Maximization

In this section, we investigate which types of predictions are useful augmentations to overcome the logarithmic barrier to approximate social welfare, the first-best benchmark (FB), which is equal to the highest value $v_1$. There are two main types of predictions considered in the learning-augmented mechanism design literature: (i) predictions about the highest value $v_1$ (that is, the first-best benchmark FB), and (ii) predictions about every agents' values (that is, the full instance). These are motivated by having black-box access to ML models that provide predictions about the current instance.

We first show that even perfect knowledge of the first best benchmark FB—that is, *how much* is the highest value $v_1$—is useless to achieve a constant-factor consistency guarantee (approximation to FB when predictions are correct). This is independent of robustness considerations. On the other hand, we find that while having predictions of each agent's value suffices, it is in fact overkill for consistency. The crucial information is the *identity* of the agent with value $v_1$, which is sufficient to achieve a constant-factor consistency guarantee, while simultaneously maintaining a constant-factor robustness guarantee (approximation to the optimal lottery $\text{OPT}_L$ when predictions are incorrect).

Our algorithm only requires a black-box binary prediction with minimal information granularity as opposed to predictions about the entire vector of values or logits. For instance, this could be an ML model that takes as input a feature vector of the current agent and outputs a binary prediction about whether or not they are the highest valued agent.

## 4.1. An Insufficient Prediction: How Much

We show that knowing *how much* the first-best benchmark (FB) is, or equivalently the value of the top agent, is insufficient: even with perfect knowledge of the value FB, no mechanism (even randomized) can achieve a constant consistency guarantee even in a relaxed offline setting. This further highlights a significant difference between utility maximization and other objectives, where a prediction of FB has been used to improve approximation guarantees (for example, for revenue in (Balkanski et al., 2024)). Our analysis closely resembles that of Qiao & Valiant (2023), who show that $o(\log n)$-consistency is impossible in pen testing, which corresponds to the restriction of using only posted price mechanisms.[4] We extend this impossibility to hold for any truthful prior-free mechanism using techniques from Bayesian mechanism design. We believe that this approach may be useful to prove consistency impossibility results in other mechanism design settings as well.

We present the theorem now together with a proof sketch. The full proof is in Appendix C.

**Theorem 3.** *Let* FB *denote the optimal social welfare. Even given the exact value of* FB*, any (prior-free) mechanism achieves at best a* $\Theta(\log n)$*-approximation to* FB*.*

*Proof sketch.* We prove the theorem by constructing a Bayesian instance where the benchmark FB is large in expectation, yet every truthful mechanism has only constant expected utility—even if it is told the realized value of FB. The distribution is chosen so that the maximum equals a fixed cap with overwhelming probability, so knowing FB provides essentially no additional actionable information. Using standard Bayesian mechanism design tools, we show that the optimal expected utility without access to FB is $O(1)$, and that the same $O(1)$ bound (up to negligible slack) holds even for mechanisms that also receive FB as input. We finally prove that the expected FB is $\Omega(\log n)$ on this instance. Thus, no mechanism (with knowledge of FB) has a constant-factor approximation to FB. Finally, if a prior-free mechanism could guarantee a constant-factor approximation given FB, applying it to this Bayesian instance and taking expectations would contradict the $O(1)$ upper bound.

□

## 4.2. A Beneficial Prediction: Who

In this section, we demonstrate that *who*, the identity of the agent who has the highest value $v_1 = $ FB, is a sufficient

prediction. Specifically, we present learning-augmented mechanisms that guarantee constant consistency and robustness when given access to online predictions. As agents arrive, the prediction at each step is a binary signal, with "yes" signaling the arrival of the agent that attains the first-best value FB. This prediction model is strictly weaker than revealing the full value profile of all agents. Moreover, the predictor doesn't reveal the identity of the top (highest-valued $v_1 = $ FB) agent in advance and only provides the binary signal online.

This alternative model of prediction provides a qualitatively different kind of information. By allowing a mechanism to identify the top agent without relying on costly screening prices, a prediction about *who* is the top agent perfectly aligns with the objective of utility maximization. We emphasize that this model of prediction is necessary in the following sense: any mechanism aiming to guarantee FB must be given enough information to identify the top agent without relying on prices to screen the agents.

We first observe that a *trivial randomized mechanism* can now easily obtain constant consistency and constant robustness simultaneously, in stark contrast to the strong impossibility when knowing *how much* the highest value is (Theorem 3). In particular, by forgoing the requirement to design a deterministic mechanism, a simple randomization (or convex combination) between "follow the prediction" and our phases mechanism $\mathcal{M}_{\mathrm{phase}}$ achieves a linear trade-off between consistency and robustness.[5]

However our main question remains: can a *deterministic mechanism* simultaneously obtain constant consistency and robustness? Further, how does it compare to the trivial convex combination obtained through randomization? We consider a class of learning-augmented mechanisms that use a prediction phase in addition to the three phases of mechanism $\mathcal{M}_{\mathrm{phase}}$. In the prediction phase, the mechanism simply follows the prediction and allocates to the predicted top agent (should they arrive) for free. A natural option for a prediction phase is at the beginning, either as a separate phase or concurrently with the sample phase. This essentially simulates the randomized mechanism, except when the prediction wrongly identifies $v_2$ as the highest valued agent (see Theorem 4).

A more interesting option is to have a prediction phase concurrently with the max phase. Intuitively, we double down to extract as much surplus possible from the highest valued agent. The max phase originally aims to capture the surplus from solely allocating to the highest valued agent (by charging a screening cost of $v_2$), so by running a prediction

---

[4]Qiao & Valiant (2023) also consider a stronger prediction model that provides a multi-set of all agents' values (without revealing which agents have which values). They show that even perfect knowledge of *how much all agents value* the item is insufficient information to obtain a constant approximation to FB.

[5]This is because a mechanism that follows the prediction allocates the item for free to predicted highest valued agent obtains 1-consistency (but 0-robustness), and on the other hand, $\mathcal{M}_{\mathrm{phase}}$ obtains $O(1)$-robustness (but $\Omega(\log n)$-consistency).

phase concurrently, we can use predictions to identify the top agent instead of subjecting the agent to a screening cost. In fact, we show even better consistency-robustness tradeoffs for this option (see Theorem 5). However, neither option ultimately beats the trivial randomized mechanism in the worst case.

### 4.2.1. BEST-OF-BOTH-WORLDS GUARANTEES

In this section, we present two ways to augment our phases mechanism with predictions. Both approaches involve defining a prediction window $P$, during which the mechanism may use the predictions. In particular, let $t_{\text{pred}}$ denote the arrival time of the predicted highest valued agent, and let $E_{\text{pred}} = \{t_{\text{pred}} \in P\}$ be the event that the predicted highest valued agent arrives during the prediction window $P$. Additionally, let $\hat{\imath}$ denote the actual rank of the predicted agent, so that $\hat{\imath} = 1$ precisely when the prediction is correct.

We will analyze these mechanisms through the familiar events $E_{max}$ and $E_{lot}$. Recall that $E_{max}$ is the event that the max-phase rule allocates the item to agent 1, at any price, and $E_{lot}$ is the event that the mechanism both reaches the lottery phase (without having allocated the item) and does so while satisfying the balanced property (previously this was guaranteed by event $E_{RO} \cap \{1 \in N_{\text{lot}} \text{ and } 2 \in N_{\text{sample}}\}$). We highlight that these events will now need to account for the prediction window event $E_{\text{pred}}$ as well.

We start by presenting the learning-augmented mechanism $\mathcal{M}_{Psamp}$ that executes the prediction window concurrently with the sample phase. This mechanism takes a prediction window length parameter $\gamma \in (0, \frac{1}{2e})$ and executes the prediction rule in $P = [1 : \gamma n]$. We prove the following theorem for mechanism $\mathcal{M}_{Psamp}$.

**Theorem 4.** *For a prediction window length parameter $\gamma \in (0, \frac{1}{2e})$, the random-order learning-augmented mechanism $\mathcal{M}_{Psamp}(\gamma)$ achieves $\gamma$-consistency and $\rho_1(\gamma)$-robustness, where $\rho_1(\gamma) \geq \frac{(1-2e\gamma)}{625e}$.*

*Proof sketch.* When the prediction is correct, that is, $\hat{\imath} = 1$, the mechanism achieves consistency only if the predicted top agent arrives in the prediction window, i.e., under event $E_{\text{pred}}$, which occurs with probability $\gamma$. Hence, the mechanism allocates to the top agent for free with probability $\gamma$, yielding consistency $\gamma$.

For robustness, we condition on $\bar{E}_{\text{pred}}$ (the predicted top agent does not arrive in the prediction window $P$), in which case the prediction rule never allocates. The analysis then largely mirrors that of $\mathcal{M}_{\text{phase}}$: for the max phase, we bound the probability that the max rule allocates to agent 1 via a standard secretary argument; for the lottery phase, we lower-bound the probability of reaching the lottery phase without allocating earlier and while the balancedness property holds.

A technical subtlety is that the latter probability depends on which agent is predicted to be highest (departing from the $\mathcal{M}_{\text{phase}}$ analysis); the worst case for our analysis occurs when $\hat{\imath} = 2$, meaning that the prediction points to the second-highest-valued agent. Accordingly, we state robustness as the minimum over the resulting subcases, and give the full guarantee in Appendix C.2. $\square$

We now formalize the learning-augmented mechanism $\mathcal{M}_{Pmax}$ that executes the prediction window concurrently with the max phase. The mechanism $\mathcal{M}_{Pmax}$ takes as input a prediction window length parameter $\alpha \in (0, \frac{e-1}{2e})$ and executes the prediction rule for $\alpha n$ steps in $P = [\frac{n}{2e} + 1 : (\frac{1}{2e} + \alpha)n]$, intersecting with the max phase. We prove the following theorem for mechanism $\mathcal{M}_{Pmax}$.

**Theorem 5.** *Let $n \geq 27$. For prediction window length $\alpha \in (0, \frac{e-1}{2e})$, the random-order learning-augmented mechanism $\mathcal{M}_{Pmax}(\alpha)$ achieves $\frac{1}{2e} \ln(1 + 2e\alpha)$-consistency and $\rho_2(\alpha)$-robustness, where $\rho_2(\alpha) \geq \frac{1-\alpha}{625e}$.*

*Proof sketch.* To prove consistency, we assume correct prediction ($\hat{\imath} = 1$) and bound the probability that $\mathcal{M}_{Pmax}$ allocates via the prediction rule. We condition on the predicted top agent arriving in the prediction window (event $E_{\text{pred}}$) and then apply a standard secretary argument to compute the allocation probability. This is formalized in the following lemma (proved in Appendix C.3).

**Lemma 4.** *When the predictor is correct, $\mathcal{M}_{Pmax}(\alpha)$ allocates to the highest valued agent for free with probability*

$$\Pr[E_{max} \cap E_{pred} \mid \hat{\imath} = 1] \geq \frac{1}{2e} \ln(1 + 2e\alpha).$$

Using this lemma, we can immediately argue that the consistency of this mechanism is $\gamma(\alpha) = \frac{1}{2e} \ln(1 + 2e\alpha)$.

We now sketch the robustness guarantee. For the max phase, we again bound the probability that the max phase allocates to agent 1, but the secretary calculation requires a minor adjustment because the prediction rule may allocate earlier and preempt agent 1 (precisely because the prediction is incorrect). We formalize and lower-bound $\Pr[E_{max}]$ in Lemma 10 (Appendix C.3). For the lottery phase, we similarly lower-bound the probability of reaching the lottery phase without allocating earlier and while the balancedness event holds. The relevant probability bounds and the resulting robustness guarantee are given in Appendix C.3. $\square$

**Comparing $\mathcal{M}_{Psamp}$ vs $\mathcal{M}_{Pmax}$.** The guarantees in Theorems 4 and 5 hold for worst case instances. However, our guarantees may be improved for specific families of instances. In fact, our analysis already provides a tighter bound in terms of $v_1 - v_2$ and $\text{OPT}_L$, see Equations (7)

and (9) in the appendix. Our two classes of learning-augmented mechanisms follow different approaches to harness predictions: $\mathcal{M}_{Pmax}$ doubles down on allocating to the highest valued agent in the max phase, but without subjecting them to a screening cost, while $\mathcal{M}_{Psamp}$ tries to simulate a convex combination (like the trivial randomization). This might suggest that the two different mechanisms could comparatively provide better or worse tradeoffs depending on the ratio of $v_1 - v_2$ and $\text{OPT}_L$.

To compare the mechanisms more directly, we first normalize the tradeoff of $\mathcal{M}_{Pmax}$ by defining $\gamma = \frac{1}{2e}\ln(1 + 2e\alpha)$ as its consistency, and deriving the robustness as a function of $\gamma$. Let $\Delta = \frac{v_1 - v_2}{\text{OPT}_L}$. The robustness guarantees of all of our mechanisms ($\mathcal{M}_{\text{phase}}$, $\mathcal{M}_{Psamp}$, $\mathcal{M}_{Pmax}$) are worst at $\Delta = 0$ and improve as $\Delta$ increases. Our bounds then show that $\mathcal{M}_{Pmax}$ achieves better robustness at the same consistency level, for any $\Delta \in [0, 1]$. The derivations appear in Appendix C.4.

Recall that a bad case for our robustness analysis of $\mathcal{M}_{Psamp}$ is when the prediction incorrectly points to the second-highest valued agent. If this is not the case, $\mathcal{M}_{Psamp}$'s guarantees are better than $\mathcal{M}_{Pmax}$. We stress that these comparisons reflect our lower-bound analysis and should not be interpreted as tight; establishing matching upper bounds is an interesting remaining open direction.

**Discussions and open directions.** We can extend the consistency-robustness tradeoffs to the entire range of $[0, 1]$ (beyond the $1/2e$ consistency in our formal theorems) with worse robustness guarantees. This requires extending the prediction window length to beyond $1/2e$ at a steeper (albeit still constant) cost to robustness.[6] Note that neither of the two deterministic mechanisms manage to ultimately beat the linear trade-off of the trivial randomized mechanism. We conjecture that more informative predictions (i.e., about all agents' values) do not provide any additional advantage. This is because any mechanism seems to have the same two options to test the predictions: it can either sample some agents to verify their predictions, or post prices based on the predictions which risks allocating at prices that are too high. We leave as an open problem whether more informative predictions about all agents' values can help beat the linear trade-off using *deterministic* mechanisms. While we establish best-of-both worlds guarantees asymptotically, our work does not provide tight guarantees nor optimizes constants, both interesting directions for future work. A natural direction for tightening our analysis is to modify the lottery phase analysis without conditioning on agent 2 appearing in the sample phase. Finally, it would be interesting to generalize the best-of-both-worlds result to be error toler-

---

[6]We need to modify the robustness analysis to only use balancedness on a smaller set of agents.

ant: for our current mechanisms, if the predictor chooses an agent who is "close" to the highest value, then our results carry over with that corresponding error, but one might consider defining different metrics to capture imperfect but not arbitrarily bad predictions.

## Impact Statement

This paper presents work whose goal is to advance the field of machine learning, mechanism design, and game theory. There are many potential societal consequences of our work, none which we feel must be specifically highlighted here.

## Acknowledgements

The work of K. Goldner and T. Tsilivis was supported by NSF CAREER Award CCF-2441071. This work was done when D. Mohan was employed at Boston University.

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

## A. Extended Related Work

**Utility Maximization.** From the perspective of utility maximization, the most relevant work to ours is the seminal work of Hartline & Roughgarden (2008), who study single-item and multi-unit auctions in the offline setting. They provide randomized mechanisms that give a constant-approximation to a prior-free benchmark (which coincides with our robustness benchmark) and an $O(1/\log n)$-approximation to the optimal welfare. Their work also considers the Bayesian mechanism design setting, which has more recently been studied beyond single item auctions in a number of recent works (Fotakis et al., 2016; Goldner & Lundy, 2025; Ezra et al., 2025; Ganesh & Hartline, 2025; Eden et al., 2026).

Qiao & Valiant (2023) consider the online pen testing problem, which is equivalent to utility maximization when restricted to posted price mechanisms. They study both the random order setting (as we do) and the Bayesian setting, providing an $O(1/\log n)$-approximation to the optimal social welfare (our consistency benchmark).

*Our work.* We adapt these two works to the random arrival setting to design a deterministic mechanism that maintains the robustness guarantee and improves the consistency guarantee to a constant-approximation.

**Predictions.** There has been a recent burst of interest in mechanism design with predictions, initiated by Agrawal et al. (2022) (on facility location) and Xu & Lu (2022) (for revenue maximization). The literature most relevant to us are those that maximize revenue. Xu & Lu (2022) consider deterministic mechanisms for revenue maximization with predictions about each agent's value. Caragiannis & Kalantzis (2024) study randomized mechanisms and provide tight consistency and robustness tradeoffs. Lu et al. (2024) consider the digital goods or $k$-unit setting ($k > 1$) and provide 1-consistency to highest value and constant robustness to a different benchmark (that doesn't apply for single item settings). Balkanski et al. (2024) consider revenue maximization in the online random order model, when having access to a prediction about the highest value. They show constant consistency to the highest value and constant robustness to second highest value.

The only prior work considering our objective of utility maximization is (Qiao & Valiant, 2023), who consider the case of having always perfect predictions.[7] Having access to the highest value prediction doesn't help improve the $1/\log n$ approximation to optimal welfare. However, having access to the multiset of all values (without knowing the identity of the agents) can improve the approximation to $\log \log n/\log n$. They do not study robustness guarantees.

The framework of algorithms with predictions and best-of-both-world guarantees originated in online algorithms. Recent work in the random order model has studied the secretary problem when there are predictions about each agent's value (Karisani et al., 2026; Fujii & Yoshida, 2024).

## B. Proofs from Section 3

### B.1. Proof of Lemma 2

**Lemma 2.** *For a uniformly random order permutation $\sigma$ and a random subset $S$ it holds that:*

$$\Pr_{\sigma}\left[E_{RO} \mid n_1 = 1, n_2 = 1\right] \geq \Pr_{S}\left[E_{RS} \mid s_1 = 1, s_2 = 1\right].$$

*Proof of Lemma 2.* Formally, we define two stochastic processes—one that partitions the instance by generating a random subset $S$, and another that generates a random permutation of the instance $\sigma$ and partitions it in the middle. To simplify the comparison we overload notation and define $N = N_{\text{lot}} = \sigma[\frac{n}{2} + 1 : n]$ and $\bar{N} = \overline{N_{\text{lot}}}$.

1. For the random subset, we define independent indicator random variables $S_i = \mathbb{1}\{i \in S\}$, and note that $\Pr[S_i = 1] = \frac{1}{2}$. The realizations of these random variables define the random subset $S$. We define as $H_i$ the realized partition of draws $1, 2, \ldots, i$ (the partition history).

2. For the random permutation, we consider the following procedure. We have $n$ empty spots on $\sigma$ to be filled. For every element $i \in \{1, 2, \ldots, n\}$, place it in any of the remaining $n + 1 - i$ empty spots uniformly at random. Define the indicator random variable $N_i = \mathbb{1}\{i \in N\}$, the realized partition $H_i$ (partition history up to $i$) as above, and notice that $\Pr[N_{i+1} = 1 | H_i] = \frac{\frac{n}{2} - n_i}{n - i}$.

---

[7]Their work is not described using the prediction framework, but for consistency guarantees, it applies. They also consider error tolerant versions, but don't provide any guarantees when the prediction is arbitrarily bad.

To enable a coupling argument, we define a series of stochastic processes $\mathcal{R}^{(i)}$, for $i = 0, 1, \ldots, n$, intermediate between the two partitioning procedures, such that $\mathcal{R}^{(0)}$ generates the random-subset partition $(S, \bar{S})$ and $\mathcal{R}^{(n)}$ generates the partition $(N, \bar{N})$ induced by splitting a uniformly random permutation in half. The process $\mathcal{R}^{(i)}$ begins by assigning the top $i$ agents according to the permutation-based partitioning rule, and the remaining $n - i$ agents according to the random-subset rule. Across processes $\mathcal{R}^{(i)}$, as $i$ increases, progressively more of the top agents are assigned according to the permutation-based rule and fewer according to the random-subset rule. This interpolation allows us to compare the two partitioning procedures draw by draw.

Formally, $\mathcal{R}^{(i)} = (R_1^{(i)}, \ldots, R_n^{(i)})$ is defined by setting $R_j^{(i)} = N_j$ for $j \leq i$ and $R_j^{(i)} = S_j$ for $j > i$, that is, $\mathcal{R}^{(i)} = (N_1^{(i)}, \ldots, N_i^{(i)}, S_{i+1}^{(i)}, \ldots, S_n^{(i)})$. Moreover, we denote $(R^{(i)}, \bar{R}^{(i)})$ to be the partition induced by process $\mathcal{R}^{(i)}$, and notice that $R^{(0)} = S$ and $R^{(n)} = N$. Let $H_k$ denote a realized partition history up to draw $k$, that is, the partition of the top $k$ agents. Notice that we suppress the dependence of this notation on a particular process $\mathcal{R}^{(i)}$ because our comparison conditions on the same realized partial history of partition $H_k$ across different processes.

For each process $\mathcal{R}^{(i)}$, let $r_j^{(i)} = \left| R^{(i)} \cap \{1, 2, \ldots, j\} \right|$ denote the number of the top $j$ agents assigned to side $R^{(i)}$ of the partition. We then define the auxiliary balancedness event for process $\mathcal{R}^{(i)}$ by $E_{\mathcal{R}^{(i)}} = \left\{ \forall j \geq 2 : \frac{1}{6}j \leq r_j^{(i)} \leq \frac{5}{6}j \right\}$. Under this notation, the inequality of Lemma 2 can be rewritten as

$$\Pr_\sigma [E_{RO} \mid n_1 = 1, n_2 = 1] \geq \Pr_S [E_{RS} \mid s_1 = 1, s_2 = 1] \Leftrightarrow$$
$$\Pr\left[ \bar{E}_{\mathcal{R}^{(0)}} \,\middle|\, r_1^{(0)} = 1, \, r_2^{(0)} = 1 \right] \geq \Pr\left[ \bar{E}_{\mathcal{R}^{(n)}} \,\middle|\, r_1^{(n)} = 1, \, r_2^{(n)} = 1 \right], \tag{4}$$

where the randomness of the second inequality is over the corresponding random processes $\mathcal{R}^{(0)}$ and $\mathcal{R}^{(n)}$ respectively and the bar notation denotes the complement of an event.

Let $H_2^*$ denote the realized history at time 2 in which the top two agents are partitioned into separate sets in the prescribed way; then the two conditioning events $\{r_1^{(0)} = 1, \, r_2^{(0)} = 1\}$ and $\{r_1^{(n)} = 1, \, r_2^{(n)} = 1\}$ both correspond to $H_2^*$.

To prove the inequality in Equation (4), we establish the following intermediate claim, which compares the probabilities of $\bar{E}_{\mathcal{R}^{(i)}}$ and $\bar{E}_{\mathcal{R}^{(i+1)}}$ after conditioning on the same realized history up to time $i$.

**Lemma 5.** *For all $1 \leq i \leq n - 1$, and any history $H_i$ we have*

$$\Pr\left[ \bar{E}_{\mathcal{R}^{(i)}} | H_i \right] \geq \Pr\left[ \bar{E}_{\mathcal{R}^{(i+1)}} | H_i \right].$$

*That is, assuming a fixed history $H_i$, the probability of the random process $\mathcal{R}^{(i)}$ being unbalanced weakly decreases in $i$.*

We defer the proof of this claim and first show how it implies Lemma 2. Let $B_{\leq t}^{(i)} = \{\forall\, 2 \leq j \leq t : \frac{1}{6}j \leq r_j^{(i)} \leq \frac{5}{6}j\}$ be the event that the partition induced by $\mathcal{R}^{(i)}$ is balanced up to element $t$. For $i \geq 3$

$$\Pr\left[ \bar{E}_{\mathcal{R}^{(i)}} | H_2^* \right] = \Pr\left[ \bar{E}_{\mathcal{R}^{(i)}} \,\middle|\, H_2^* \cap B_{\leq i-1}^{(i)} \right] \Pr\left[ B_{\leq i-1}^{(i)} \,\middle|\, H_2^* \right] + \Pr\left[ \bar{B}_{i-1}^{(i)} \,\middle|\, H_2^* \right]$$
$$\leq \Pr\left[ \bar{E}_{\mathcal{R}^{(i-1)}} \,\middle|\, H_2^* \cap B_{\leq i-1}^{(i-1)} \right] \Pr\left[ B_{\leq i-1}^{(i-1)} \,\middle|\, H_2^* \right] + \Pr\left[ \bar{B}_{i-1}^{(i-1)} \,\middle|\, H_2^* \right]$$
$$= \Pr\left[ \bar{E}_{\mathcal{R}^{(i-1)}} | H_2^* \right], \tag{5}$$

where the equalities follow by observing that if a random process is unbalanced, then either it only becomes unbalanced after assigning element $i - 1$, or was already unbalanced before element $i$, and the inequality follows by applying Lemma 5 pointwise for all histories in $H_2^* \cap B_{\leq i-1}^{(i)}$.

Recursively applying Equation (5) for $i = 3, \ldots, n$, we have:

$$\Pr[\bar{E}_{\mathcal{R}^{(0)}} | H_2^*] = \Pr[\bar{E}_{\mathcal{R}^{(1)}} | H_2^*] = \Pr[\bar{E}_{\mathcal{R}^{(2)}} | H_2^*] \geq \Pr[\bar{E}_{\mathcal{R}^{(3)}} | H_2^*] \geq \cdots \geq \Pr[\bar{E}_{\mathcal{R}^{(n)}} | H_2^*],$$

where the first two equalities are because processes $\mathcal{R}^{(0)}$, $\mathcal{R}^{(1)}$, and $\mathcal{R}^{(2)}$ coincide if you fix the first two draws. $\qquad \square$

We now prove Lemma 5.

*Proof of Lemma 5.* If $H_i$ is such that the balanced property doesn't hold for some $i' \leq i$, then the claim trivially holds since both probabilities are 1. Suppose from now on that up to $i$, the balanced property holds.

Observe that under event $\bar{E}_{\mathcal{R}^{(i)}}$, there exists some $t \geq 2$ where the balancedness becomes violated in process $\mathcal{R}^{(i)}$, i.e., $\min(r_t^{(i)} - \frac{t}{6}, \frac{5t}{6} - r_t^{(i)}) < 0$. We can equivalently measure the quantity $\min(\frac{5t}{6} - \bar{r}_t^{(i)}, \frac{5t}{6} - r_t^{(i)})$ to know how close the process $\mathcal{R}^{(i)}$ is to violating the upper boundary of balancedness (by either of the two sides of the partition), where $\bar{r}_t^{(i)} = t - r_t^{(i)}$.

In particular, to keep track of this quantity, we consider a random walk on the non-negative integers $\{0, 1, 2 \dots\}$. The state of the random walk at time $j$ (defined as state$[j]$) will capture how close the larger side of the process is to violating the upper boundary of balancedness. The walk moves at each time exactly one step (either to the left or to the right), with a right step at time $j$ corresponding to adding the $j$-th element to the larger side of the partition. The walk is initialized at $0$, meaning state$[0] = 0$. We define the transition probabilities of this random walk appropriately for process $\mathcal{R}^{(i)}$, that is:

- If state$[j] = 0$ then state$[j+1] = 1$ with probability $1$.[8]

- If state$[j] \neq 0$, then:
    - for steps at time $j < i$, the probability of moving to the right is $\min\{p_j, 1 - p_j\}$, where $p_j := \Pr[N_{j+1} = 1 | H_j]$.
    - for steps at time $j \geq i$, the probability of moving to the right is $\frac{1}{2}$.

Notice now for $\mathcal{R}^{(i)}$ that, assuming a given history $H_j$, the equivalent random walk is at state$[j] = |2r_j^{(i)} - j|$ (which is exactly the difference between the two sides of the partition). Finally, note that reaching the boundary of balancedness (i.e., $\frac{5t}{6}$) corresponds to the state of $\frac{2t}{3}$ under the random walk model and $E_{\mathcal{R}^{(i)}} \equiv \{\forall j \geq 2 : \text{state}[j] \leq \frac{2j}{3}\}$.

Returning to the proof, suppose now that $H_i$ corresponds to the random walk having state$[i] = k$, and recall that $k \geq 0$. If $k = 0$, then the claim trivially holds as the processes coincide for draws $i+2, \dots, n$ (by definition) and for draw $i+1$ as well (the sides are equal up to $i$ so we have $\Pr(N_{i+1} = 1 | H_i) = 1/2$, meaning that $N_{i+1} \equiv S_{i+1}$). If $k > 0$, we have:

$$
\begin{aligned}
\Pr\left[\bar{E}_{\mathcal{R}^{(i)}} | \text{state}[i] = k\right] &= \Pr\left[\bar{E}_{\mathcal{R}^{(i)}} | \text{state}[i+1] = k-1\right] \cdot \frac{1}{2} \\
&\quad + \Pr\left[\bar{E}_{\mathcal{R}^{(i)}} | \text{state}[i+1] = k+1\right] \cdot \frac{1}{2} \\
&\geq \Pr\left[\bar{E}_{\mathcal{R}^{(i)}} | \text{state}[i+1] = k-1\right] \cdot \max\{p_i, 1 - p_i\} \\
&\quad + \Pr\left[\bar{E}_{\mathcal{R}^{(i)}} | \text{state}[i+1] = k+1\right] \cdot \min\{p_i, 1 - p_i\} \\
&= \Pr\left[\bar{E}_{\mathcal{R}^{(i+1)}} | \text{state}[i] = k\right],
\end{aligned}
\tag{6}
$$

where the inequality is because:

1. $\Pr\left[\bar{E}_{\mathcal{R}^{(i)}} | \text{state}[i+1] = k+1\right] \geq \Pr\left[\bar{E}_{\mathcal{R}^{(i)}} | \text{state}[i+1] = k-1\right]$ because the probability of becoming unbalanced is larger the closer you are to the boundary (to the right).

2. Trivially, $\max\{p_i, 1 - p_i\} > \frac{1}{2} > \min\{p_i, 1 - p_i\}$.

3. Moving from the fair coin $S_{i+1}$ to the biased coin $N_{i+1}$ shifts probability mass toward the $k - 1$ state, which is precisely the state with the lower conditional probability of imbalance.

Thus, for any history $H_i$, we have argued:

$$
\Pr\left[\bar{E}_{\mathcal{R}^{(i)}} | H_i\right] \geq \Pr\left[\bar{E}_{\mathcal{R}^{(i+1)}} | H_i\right].
$$

$\square$

---

[8]When both sides are equal, any element added will be to the resulting larger side.

## B.2. Parallel Analysis to (Hartline & Roughgarden, 2008) for Online Partitioning

The analysis in this section is parallel to that of (Hartline & Roughgarden, 2008) for RSOL, but modified for our online random-order (RO) setting rather than their offline random-sampling (RS) setting. We restate and prove Lemma 3.

**Lemma 3.** *Conditioning on event $E_{lot}$, the expected utility of $\mathcal{M}_{\text{phase}}$ is at least $\frac{\text{OPT}_L - (v_1 - v_2)}{125}$.*

*Proof.* We start by arguing that under event $E_{lot} = E_{RO} \cap \{1 \in N_{\text{lot}}, 2 \in N_{\text{sample}}\}$., the expected utilities of any posted price $p$ on $\overline{N_{\text{lot}}}$ and on $N_{\text{lot}}$ are within a constant factor.

**Lemma 6.** *Assuming event $E_{lot}$, the expected utility of any posted price $p$ on $N_{lot}$ is a $\frac{1}{25}$-approximation to the expected utility of the same posted price $p$ on $\overline{N_{lot}}$.*

*Proof.* We again leverage the equivalence of an online posted-price and a $p$-lottery. Let $U(T, p)$ denote the expected utility of a $p$-lottery on a set of agents $T$. We express the expected utility of a $v_{k+1}$-lottery as follows (see figure 1 for a proof by picture):

$$U(T, v_{k+1}) = \frac{1}{|T \cap \{1, \ldots, k\}|} \left( \sum_{i \in T \cap \{1, 2, \ldots, k\}} v_i \right) - v_{k+1} = \frac{1}{|T \cap \{1, \ldots, k\}|} \sum_{i \leq k} |T \cap \{1, \ldots, i\}| \cdot (v_i - v_{i+1}).$$

Consider now any $p$-lottery on $N_{\text{lot}}$ posting a price of $v_{k+1}$. Notice that since we have assumed event $E_{lot} = E_{RO} \cap \{1 \in N_{\text{lot}}, 2 \in N_{\text{sample}}\}$ occurs, we have for $i \geq 2$ that both $n_i, \bar{n}_i$ are in $[\frac{i}{6}, \frac{5i}{6}]$, or equivalently, $n_i \in [\frac{\bar{n}_i}{5}, 5\bar{n}_i]$ (and $\bar{n}_1 = 0$ and $n_1 = n_2 = \bar{n}_2 = 1$). We lower bound the expected utility of posting $v_{k+1}$ on $N_{\text{lot}}$:

$$U(N_{\text{lot}}, v_{k+1}) = \frac{1}{n_k} \sum_{i=1}^{k} n_i(v_i - v_{i+1}) \geq \frac{1}{5\bar{n}_k} \sum_{i=1}^{k} \frac{1}{5} \bar{n}_i(v_i - v_{i+1}) = \frac{1}{25} U(\overline{N_{\text{lot}}}, v_{k+1}).$$

$\square$

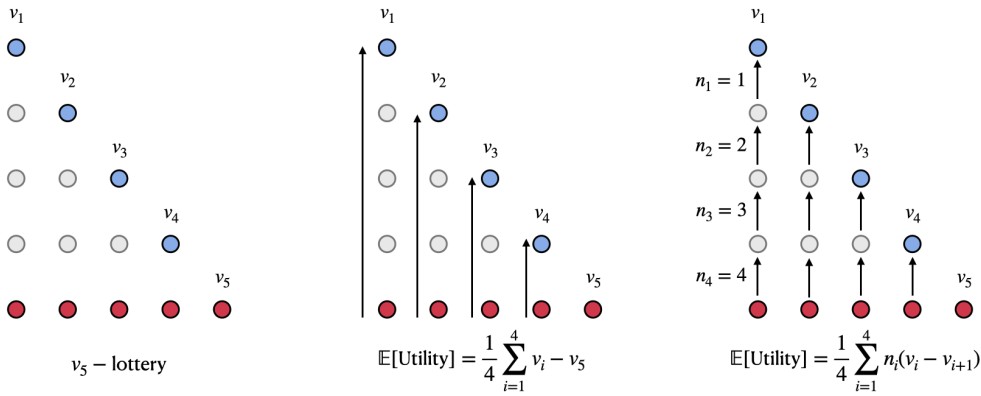

*Figure 1.* Expected utility of a $v_5$-lottery.

We continue by arguing that the expected utility of the optimal $p$-lottery in $\overline{N_{\text{lot}}}$ is approximately the expected utility of the optimal $p$-lottery over all agents $\text{OPT}_L$.

**Lemma 7.** *Assuming events $E_{lot}$, the expected utility of the optimal $p$-lottery in $\overline{N_{lot}}$ is a $\frac{1}{5}$-approximation to $\text{OPT}_L - (v_1 - v_2)$.*

*Proof.* Assume that the optimal $p$-lottery on $\overline{N_{\text{lot}}}$ is a price of $v_{k+1}$, and the optimal $p$-lottery in the entire instance is $v_{l^*+1}$. We bound the expected utility of the optimal lottery on $\overline{N_{\text{lot}}}$:

$$U(\overline{N_{\text{lot}}}, v_{k+1}) \geq U(\overline{N_{\text{lot}}}, v_{l^*+1}) = \frac{1}{\bar{n}_{l^*}} \sum_{i=1}^{l^*} \bar{n}_i(v_i - v_{i+1}) \geq \frac{6}{5l^*} \sum_{i=2}^{l^*} \frac{i}{6}(v_i - v_{i+1}) \geq \frac{\text{OPT}_L - (v_1 - v_2)}{5},$$

where the first inequality is by optimality of $v_{k+1}$ on the $\overline{N_{\text{lot}}}$ and the second inequality follows because $\bar{n}_i \geq i/6$ and $\bar{n}_{l^*} \leq \frac{5}{6}l^*$ by $E_{RO}$. $\qquad\square$

Putting things together, we can lower bound the expected utility of the $v_{k+1}$-lottery on $N_{\text{lot}}$ (where $v_{k+1}$ is the optimal lottery in $\overline{N_{\text{lot}}}$)

$$U(N_{\text{lot}}, v_{k+1}) \geq \frac{1}{25} U(\overline{N_{\text{lot}}}, v_{k+1}) \geq \frac{1}{125} \left(\text{OPT}_L - (v_1 - v_2)\right),$$

where we have used Lemma 6 for the first inequality and Lemma 7 for the second. $\qquad\square$

## C. Proofs from Section 4

### C.1. Missing proof from Subsection 4.1

**Theorem 3.** *Let* FB *denote the optimal social welfare. Even given the exact value of* FB*, any (prior-free) mechanism achieves at best a* $\Theta(\log n)$*-approximation to* FB*.*

*Proof.* To prove the theorem, we show that for any mechanism that knows the value of FB, there exists a Bayesian instance where this mechanism's expected utility is $O(1)$, while the expected first-best benchmark is $\Omega(\log n)$.

We consider two classes of mechanisms: $\mathcal{C}$, which only takes agent bids as input, and $\mathcal{C}'$, which takes both bids and the *true* numerical value of FB as inputs. Our goal is to show that the optimal mechanism in $\mathcal{C}'$, which *knows* FB, is upper-bounded by a mechanism in $\mathcal{C}$, which does not.

Let $U(\mathcal{M})$ denote the expected utility of a mechanism $\mathcal{M}$. We define a Bayesian instance $Y$ to compare the two mechanisms. In this instance, each agent $i$ has value $v_i$ drawn independently from the truncated exponential distribution $Y_i = \min\left(X_i, \frac{\ln n}{2}\right)$ where $X_i \sim \text{Exp}(1)$. Note that the maximum possible value is $\frac{\ln n}{2}$, and the probability that this value is realized by at least one agent can be lower-bounded as:

$$\begin{aligned}
\Pr\left[\text{FB} = \tfrac{\ln n}{2}\right] &= \Pr\left[\exists i : Y_i = \tfrac{\ln n}{2}\right] \\
&= 1 - \left(1 - e^{-\frac{\ln n}{2}}\right)^n \\
&= 1 - \left(1 - \tfrac{1}{\sqrt{n}}\right)^n \geq 1 - e^{-\sqrt{n}}.
\end{aligned}$$

Thus, with high probability, the realized value of FB equals $\frac{\ln n}{2}$, and thus if we set $n$ to be large, knowledge of FB becomes redundant. Additionally, in this instance, the benchmark we compare against is the expected maximum value, which we can lower bound as:

$$\mathbb{E}[\max_i (Y_i)] \geq \left(1 - e^{-\sqrt{n}}\right) \cdot \tfrac{\ln n}{2} = \Omega(\log n).$$

We start by analyzing the optimal mechanism OPT in $\mathcal{C}$ (i.e., without knowledge of FB). The distribution of each agent's values and their virtual utility are:

$$f_Y(y) = \begin{cases} e^{-y}, & 0 < y < \frac{\ln n}{2}, \\ \frac{1}{\sqrt{n}}, & y = \frac{\ln n}{2}, \end{cases}$$

and $\vartheta(y) = \mathbb{1}[y < \frac{\ln n}{2}]$, which dictates that the distribution satisfies the MHR condition.

The optimal mechanism for i.i.d. agents with MHR distributions is a 0-lottery (Corollary 2.11 of (Hartline & Roughgarden, 2008)). We now compute its expected value, which is simply the expected value of any draw from the distribution $Y_i$:

$$\begin{aligned}
U(\text{OPT}) = \mathbb{E}[Y_i] &= \int_0^\infty \Pr(\min(X, \tfrac{\ln n}{2}) > t)\, dt \\
&= \int_0^{\frac{\ln n}{2}} e^{-t}\, dt \\
&= 1 - \frac{1}{\sqrt{n}} = O(1).
\end{aligned}$$

Consider now $\text{OPT}'(\text{FB})$, the optimal randomized mechanism for this instance that additionally knows the realized value of FB. To upper-bound its performance, we compare $\text{OPT}'(\frac{\ln n}{2})$ to OPT; that is, we consider the mechanism $\text{OPT}'$ when it is run with the fixed (and potentially incorrect) input that FB is $\frac{\ln n}{2}$. We have:

$$
\mathbb{E}\big[U\big(\text{OPT}'(\tfrac{\ln n}{2})\big) \mid \text{FB} = \tfrac{\ln n}{2}\big]
$$
$$
= \frac{U\big(\text{OPT}'(\tfrac{\ln n}{2})\big)}{\Pr\big[\text{FB} = \tfrac{\ln n}{2}\big]}
$$
$$
\quad - \frac{\mathbb{E}\big[U\big(\text{OPT}'(\tfrac{\ln n}{2})\big) \mid \text{FB} < \tfrac{\ln n}{2}\big]\Pr\big[\text{FB} < \tfrac{\ln n}{2}\big]}{\Pr\big[\text{FB} = \tfrac{\ln n}{2}\big]}
$$
$$
\leq \frac{U(\text{OPT})}{\Pr\big[\text{FB} = \tfrac{\ln n}{2}\big]},
$$

where the equality follows from the law of total expectation, and the inequality uses non-negativity of utility (from IR) together with the optimality of OPT in class $\mathcal{C}$ (among mechanisms that do not know the true FB).

We can now upper bound the expected utility of $\text{OPT}'$ when it is given the true value of FB:

$$
U(\text{OPT}'(\text{FB}))
$$
$$
= \mathbb{E}\big[U(\text{OPT}'(\tfrac{\ln n}{2})) \mid \text{FB} = \tfrac{\ln n}{2}\big]\Pr\big[\text{FB} = \tfrac{\ln n}{2}\big]
$$
$$
\quad + \mathbb{E}\big[U(\text{OPT}'(\text{FB})) \mid \text{FB} < \tfrac{\ln n}{2}\big]\Pr\big[\text{FB} < \tfrac{\ln n}{2}\big]
$$
$$
\leq U(\text{OPT}) + \tfrac{\ln n}{2} \cdot e^{-\sqrt{n}} = O(1),
$$

where the inequality comes from the expression above as well as trivially upper bounding the utility with $\frac{\ln n}{2}$ in the exponentially low probability event of $FB < \frac{\ln n}{2}$.

Hence any Bayesian mechanism $\mathcal{M}'$ that knows the value of FB has expected utility on this instance of at most $O(1)$.

We now extend this impossibility to *prior-free* mechanisms. Suppose, for contradiction, that there exists a prior-free mechanism $\mathcal{A}$ which, given FB, achieves a constant-factor approximation to FB on every valuation profile. Apply $\mathcal{A}$ to the Bayesian instance defined above and take expectations over the randomness of the instance. By the assumed prior-free guarantee, we must have $\mathbb{E}[U(\mathcal{A})] \geq c \cdot \mathbb{E}[\text{FB}]$, for some constant $c > 0$. This, however, is a contradiction.

Thus, no prior-free mechanism with access to the value of FB can achieve a constant-factor approximation to FB. $\qquad\square$

## C.2. Missing proofs in the analysis of $\mathcal{M}_{Psamp}$.

We recall the theorem describing the mechanism's $\mathcal{M}_{Psamp}$ guarantees.

**Theorem 4.** *For a prediction window length parameter $\gamma \in (0, \frac{1}{2e})$, the random-order learning-augmented mechanism $\mathcal{M}_{Psamp}(\gamma)$ achieves $\gamma$-consistency and $\rho_1(\gamma)$-robustness, where $\rho_1(\gamma) \geq \frac{(1-2e\gamma)}{625e}$.*

*Proof.* In the main body, we concluded on the consistency of the mechanism, so here we analyze the robustness guarantee.

We start with the max phase. For mechanism $\mathcal{M}_{Psamp}$, the favorable event for this phase is $E_{max}$, defined as $E_{max} = E_{sec} \cap \bar{E}_{pred} \cap \{1 \in \overline{N_{lot}}\}$. Recall that event $E_{sec}$ (which corresponds to the secretary analysis) is independent of both event $\bar{E}_{pred}$ and event $\{1 \in \overline{N_{lot}}\}$. We prove the following lemma that ties to the calculation of $\Pr[E_{max}]$.

**Lemma 8.** *Let $n \geq 2$ and let $\sigma$ be a uniformly random permutation of $[n]$. Let the prediction window be $P = [1 : \gamma n]$ for some $\gamma \in (0, \frac{1}{2e})$ Then*

$$
\Pr[\bar{E}_{pred} \cap 1 \in \overline{N_{lot}}] = \frac{1 - \gamma}{2} + \frac{\gamma}{2(n-1)} \geq \frac{1 - \gamma}{2}
$$

PROOF OF LEMMA 8.

We lower bound the following probability:

$$
\begin{aligned}
\Pr[\bar{E}_{\text{pred}} \cap 1 \in \overline{N_{\text{lot}}}] &= \Pr[1 \in \overline{N_{\text{lot}}} \mid t_{\text{pred}} \in \overline{N_{\text{lot}}} \setminus P] \cdot \Pr[t_{\text{pred}} \in \overline{N_{\text{lot}}} \setminus P] \\
&\quad + \Pr[1 \in \overline{N_{\text{lot}}} \mid t_{\text{pred}} \in N_{\text{lot}}] \cdot \Pr[t_{\text{pred}} \in N_{\text{lot}}] \\
&= \frac{\frac{n}{2} - 1}{n - 1} \cdot \left(\frac{1}{2} - \gamma\right) + \frac{\frac{n}{2}}{(n-1)} \cdot \frac{1}{2} \\
&= \frac{n - \gamma n + 2\gamma - 1}{2(n-1)} \\
&= \frac{1 - \gamma}{2} + \frac{\gamma}{2(n-1)} \\
&\geq \frac{1 - \gamma}{2}.
\end{aligned}
$$

$\square$

We can now lower bound the probability of $E_{max}$:

$$
\Pr[E_{max}] = \Pr[E_{sec}] \cdot \Pr[\bar{E}_{\text{pred}} \cap 1 \in \overline{N_{\text{lot}}}] \geq \frac{1 - \gamma}{2e}.
$$

Finally, note that $E_{max}$ is the event that we allocate to agent 1 during the max phase, and that this guarantees utility of at least $v_1 - v_2$.

Now for the lottery phase, we can lower-bound the expected utility under event $E_{lot}$, which is essentially the event that the mechanism reaches the lottery phase, without having allocated the item, and with the balancedness property holding. For mechanism $\mathcal{M}_{Psamp}$, this event is defined as $E_{lot} = E_{RO} \cap \bar{E}_{\text{pred}} \cap \{1 \in N_{\text{lot}} \text{ and } 2 \in N_{\text{sample}}\}$, which, besides the balancedness event $E_{RO}$, boils down to computing the arrival probabilities of three agents. The subtlety in the analysis of the lottery phase has to do with whether or not the prediction incorrectly signals 2 as the top agent rather than 1. We consider now these cases:

**Case 1. Predicted max is not the second-highest agent.**   We start by proving the following useful lemma.

**Lemma 9.** *Let $n \geq 3$ and let $\sigma$ be a uniformly random permutation of $[n]$. Let the prediction window be $P = [1 : \gamma n]$ for some $\gamma \in (0, \frac{1}{2e})$. Assume the predicted highest agent is neither agent 1 nor agent 2, that is $\hat{\imath} \notin \{1, 2\}$. Then*

$$
\Pr[\bar{E}_{pred} \cap \{1 \in N_{lot} \text{ and } 2 \in N_{sample}\} \mid \hat{\imath} \notin \{1, 2\}] = \frac{n\big(n(1 - \gamma) + 2e\gamma - 2\big)}{4e\,(n-1)(n-2)} \geq \frac{1 - \gamma}{4e}.
$$

*Proof.* We prove:

$$
\begin{aligned}
\Pr[\bar{E}_{\text{pred}} \cap \{1 \in N_{\text{lot}} \text{ and } 2 \in N_{\text{sample}}\} \mid \hat{\imath} \notin \{1, 2\}] &= \frac{1}{2}\Pr[\bar{E}_{\text{pred}} \cap 2 \in N_{\text{sample}} \mid 1 \in N_{\text{lot}} \cap \hat{\imath} \notin \{1, 2\}] \\
&= \frac{1}{2}\left(\frac{\gamma n}{n - 1} \cdot \frac{n - \gamma n - 1}{n - 2} + \frac{(\frac{1}{2e} - \gamma)n}{n - 1} \cdot \frac{n - \gamma n - 2}{n - 2}\right) \\
&= \frac{n(n(1 - \gamma) + 2e\gamma - 2)}{4e(n-1)(n-2)} \\
&= \frac{1 - \gamma}{4e} + \frac{n - 2 + \gamma\big((2e - 3)n + 2\big)}{4e\,(n-1)(n-2)} \\
&\geq \frac{1 - \gamma}{4e},
\end{aligned}
$$

for $n \geq 3$.

$\square$

We can now lower bound the probability of $E_{lot}$:

$$\Pr[E_{lot} \mid \hat{\imath} \notin \{1,2\}] = \Pr[E_{RO} \mid \bar{E}_{\text{pred}} \cap \{1 \in N_{\text{lot}} \text{ and } 2 \in N_{\text{sample}}\} \cap \hat{\imath} \notin \{1,2\}]$$
$$\cdot \Pr[\bar{E}_{\text{pred}} \cap \{1 \in N_{\text{lot}} \text{ and } 2 \in N_{\text{sample}}\} \mid \hat{\imath} \notin \{1,2\}]$$
$$\geq \frac{4}{5} \cdot \frac{1-\gamma}{4e} = \frac{1-\gamma}{5e}.$$

With this in hand, we can compute the expected utility $l_1$ of the lottery phase as:

$$l_1 = \mathbb{E}[\text{Utility of lottery phase of } \mathcal{M}_{Psamp} \mid \hat{\imath} \notin \{1,2\}]$$
$$\geq \mathbb{E}[\text{Utility of lottery phase of } \mathcal{M}_{Psamp} \mid E_{lot} \cap \hat{\imath} \notin \{1,2\}] \cdot \Pr[E_{lot} \mid \hat{\imath} \notin \{1,2\}]$$
$$= \frac{\text{OPT}_L - (v_1 - v_2)}{5} \cdot \frac{1}{25} \cdot \frac{1-\gamma}{5e}$$
$$= (1-\gamma)\frac{\text{OPT}_L - (v_1 - v_2)}{625e},$$

where we have used Lemmas 6 and 7 for the utility guarantees.

Notice that if we combine this guarantee with the max phase guarantee, we get that for this subcase the mechanism achieves $(1 - \gamma)$ times the robustness guarantee of mechanism $\mathcal{M}_{\text{phase}}$. As such, under this case, it is essentially equivalent to coin-tossing between $\mathcal{M}_{\text{phase}}$ and a 1-consistent mechanism.

**Case 2. Predicted max is the second-highest agent.** Suppose now the predicted top agent is 2, that is, $\hat{\imath} = 2$. Recall that the guarantee for the lottery phase hinges on the events that i) the balanced property holds, ii) agent 1 is in the lottery phase, and iii) agent 2 is in the sample phase. Importantly, since agent 2 is the predicted top agent, the probability of the intersection of events $\bar{E}_{\text{pred}} \cap \{1 \in N_{\text{lot}} \text{ and } 2 \in N_{\text{sample}}\}$ is much smaller than in the previous case. Specifically, we compute:

$$\Pr[E_{lot} \mid \hat{\imath} = 2] = \Pr[E_{RO} \mid \bar{E}_{\text{pred}} \cap \{1 \in N_{\text{lot}} \text{ and } 2 \in N_{\text{sample}}\} \cap \hat{\imath} = 2]$$
$$\cdot \Pr[\bar{E}_{\text{pred}} \cap \{1 \in N_{\text{lot}} \text{ and } 2 \in N_{\text{sample}}\} \mid \hat{\imath} = 2]$$
$$\geq \frac{4}{5} \cdot \frac{1}{2}\left(\frac{1}{2e} - \gamma\right) = \frac{1 - 2e\gamma}{5e}.$$

With this in hand, we can compute the expected utility $l_2$ of the lottery phase (similarly to above and Equation (3)):

$$l_2 = \mathbb{E}[\text{Utility of lottery phase of } \mathcal{M}_{Psamp} \mid \hat{\imath} = 2]$$
$$\geq \mathbb{E}[\text{Utility of lottery phase of } \mathcal{M}_{Psamp} \mid E_{lot} \cap \hat{\imath} = 2] \cdot \Pr[E_{lot} \mid \hat{\imath} = 2]$$
$$= \frac{\text{OPT}_L - (v_1 - v_2)}{5} \cdot \frac{1}{25} \cdot \frac{1 - 2e\gamma}{5e}$$
$$= (1 - 2e\gamma)\frac{\text{OPT}_L - (v_1 - v_2)}{625e},$$

where we have used Lemmas 6 and 7 for the utility guarantees.

**Final Robustness Guarantee.** The robustness guarantee must be written as the minimum of the guarantees of the two subcases, that is $\min(l_1, l_2)$. From the expressions, it is obvious that $l_2$ is actually always the smallest, and with this, we lower bound the expected utility of mechanism $\mathcal{M}_{Psamp}$.

$$\mathbb{E}[\text{Utility of } \mathcal{M}_{Psamp}] \geq \mathbb{E}[\text{Utility of } \mathcal{M}_{Psamp} | E_{max}] \cdot \Pr[E_{max}] + \mathbb{E}[\text{Utility of } \mathcal{M}_{Psamp} | E_{lot}] \cdot \Pr[E_{lot}]$$
$$\geq (v_1 - v_2) \cdot \frac{1-\gamma}{2e} + (1 - 2e\gamma)\frac{\text{OPT}_L - (v_1 - v_2)}{625e} \tag{7}$$
$$= \frac{2(1 - 2e\gamma) \cdot \text{OPT}_L + (623 - (625 - 4e)\gamma) \cdot (v_1 - v_2)}{1250e}.$$

We have thus expressed the robustness guarantee as a function of only $\gamma$.

$\square$

## C.3. Missing proofs in the analysis of $\mathcal{M}_{Pmax}$.

We recall the theorem describing the mechanism's $\mathcal{M}_{Pmax}$ guarantees.

**Theorem 5.** *Let $n \geq 27$. For prediction window length $\alpha \in (0, \frac{e-1}{2e})$, the random-order learning-augmented mechanism $\mathcal{M}_{Pmax}(\alpha)$ achieves $\frac{1}{2e} \ln(1 + 2e\alpha)$-consistency and $\rho_2(\alpha)$-robustness, where $\rho_2(\alpha) \geq \frac{1-\alpha}{625e}$.*

*Proof.* We set up some auxiliary notation. Let $s = \frac{n}{2e}$ denote the length of the sample phase, and let $m = \alpha n$ be the number of steps of the prediction rule. We restate and prove Lemma 4, which we used to argue the consistency guarantee of the mechanism.

**Lemma 4.** *When the predictor is correct, $\mathcal{M}_{Pmax}(\alpha)$ allocates to the highest valued agent for free with probability*

$$\Pr\left[E_{max} \cap E_{pred} \mid \hat{\imath} = 1\right] \geq \frac{1}{2e} \ln(1 + 2e\alpha).$$

*Proof.* Fix the arrival position of the predicted highest agent $t_{\text{pred}}$ to be $t \in P = \{s+1, \ldots, s+m\}$. The mechanism allocates to this agent iff the maximum among the first $t-1$ arrivals lies in positions $\{1, 2, \ldots, s\}$. Since the arrival order is uniformly random, the index of the maximum among the first $t-1$ arrivals is uniformly distributed over $\{1, \ldots, t-1\}$, and therefore

$$\Pr\left[E_{max} \mid t_{\text{pred}} = t, \, \hat{\imath} = 1\right] = \frac{s}{t-1}.$$

Conditioned on $E_{pred}$ and $\hat{\imath} = 1$, the arrival position $t_{\text{pred}}$ is uniform over $P$ (recall that $|P| = m$). Hence,

$$\Pr[E_{max} \mid E_{\text{pred}}, \, \hat{\imath} = 1] = \frac{1}{m} \sum_{t=s+1}^{s+m} \frac{s}{t-1} = \frac{s}{m}\left(H_{s+m-1} - H_{s-1}\right).$$

We explain how to lower bound the difference $H_{s+m-1} - H_{s-1} = \sum_{t=s}^{s+m-1} \frac{1}{t}$. Using the standard integral comparison for decreasing functions, we argue that

$$\int_s^{s+m} \frac{dx}{x} \leq \sum_{t=s}^{s+m-1} \frac{1}{t} \quad \Rightarrow \quad \ln\left(\frac{s+m}{s}\right) \leq H_{s+m-1} - H_{s-1} \tag{8}$$

As such, we can lower bound the probability by:

$$\begin{aligned} \Pr[E_{max} \cap E_{\text{pred}} \mid \hat{\imath} = 1] &= \Pr[E_{max} \mid E_{\text{pred}}, \, \hat{\imath} = 1] \cdot \Pr[E_{\text{pred}} \mid \hat{\imath} = 1] \\ &\geq \frac{s}{m} \ln\left(\frac{s+m}{s}\right) \cdot \frac{m}{n} \\ &= \frac{1}{2e} \ln(1 + 2e\alpha). \end{aligned}$$

$\square$

We proceed now to the robustness guarantee. We omit conditioning on $\hat{\imath} \neq 1$ as it does not alter any of the arguments. As we have discussed in the main body, we will have to slightly alter the analysis of the max-phase. We compute the probability of event $E_{max}$ with the following lemma.

**Lemma 10.** *When the predictor is incorrect, mechanism $\mathcal{M}_{Pmax}(\alpha)$ allocates to the highest value agent $v_1$ during the max phase with probability*

$$\Pr[E_{max}] \geq \frac{1}{2e(n-1)}\left((n-1)(1-2\alpha) + \left(\alpha n - 1 + \frac{n}{2e}\right)\ln(1+2e\alpha)\right).$$

*Proof.* For the max phase, as before, we quantify the probability that the agent with value $v_1$ is allocated during the max phase. To this end, we first condition on the highest value agent being in $\overline{N_{\text{lot}}}$, as before. We decompose the probability that we allocate to the highest value agent, subject to whether or not event $E_{\text{pred}}$ is realized.

If event $E_{\text{pred}}$ is realized, we need to account for the prediction rule not executing during the max phase. It is convenient to also condition on the event that agent 1 also lies in $P$, which happens with probability:

$$\Pr[1 \in P | t_{\text{pred}} \in P \cap 1 \in \overline{N_{\text{lot}}}] = \frac{m-1}{\frac{n}{2} - 1}$$

Condition now on the event that agent 1 arrives at the exact position $j \in P$. The mechanism allocates to them if:

- The maximum among the first $j-1$ arrivals lies in positions $[1:s]$, denoted as $S_j$ (this is the usual secretary condition),
- The predicted agent $t_{\text{pred}}$ appears after position $j$, that is $t_{\text{pred}} > j$.

These two conditions give

$$\Pr[E_{max} \mid 1 \text{ at position } j, t_{\text{pred}} \in P] = \Pr[S_j] \cdot \Pr[t_{\text{pred}} > j \mid t_{\text{pred}} \in P \cap 1 \text{ at position } j]$$

$$= \frac{s}{j-1} \cdot \frac{m - (j-s)}{m-1}.$$

Averaging over the position $j$ of agent 1 within $P$ yields the exact expression

$$\Pr[E_{max} \mid 1 \in P, t_{\text{pred}} \in P] = \frac{1}{m} \sum_{j=s+1}^{s+m} \frac{s}{j-1} \cdot \frac{m-(j-s)}{m-1}$$

$$= \frac{s}{m(m-1)} \sum_{j=s+1}^{s+m} \left( \frac{s+m-1}{j-1} - 1 \right)$$

$$= \frac{s}{m(m-1)} \Big( (s+m-1)\big(H_{s+m-1} - H_{s-1}\big) - m \Big).$$

We conclude that:

$$\Pr[E_{max} \mid t_{\text{pred}} \in P \cap 1 \in \overline{N_{\text{lot}}}] = \Pr[E_{max} \mid 1 \in P, t_{\text{pred}} \in P] \cdot \Pr[1 \in P \mid t_{\text{pred}} \in P \cap 1 \in \overline{N_{\text{lot}}}]$$

$$= \frac{s}{m(m-1)} \Big( (s+m-1)\big(H_{s+m-1} - H_{s-1}\big) - m \Big) \cdot \frac{m-1}{\frac{n}{2} - 1}$$

$$\geq \frac{1}{\alpha e(n-2)} \left( \left( \frac{n}{2e} + \alpha n - 1 \right) \ln(1 + 2e\alpha) - \alpha n \right).$$

where we have substituted the harmonic difference by Equation (8).

Notice now that the corresponding probability is:

$$\Pr[t_{\text{pred}} \in P \cap 1 \in \overline{N_{\text{lot}}}] = \Pr[t_{\text{pred}} \in P \mid 1 \in \overline{N_{\text{lot}}}] \cdot \Pr[1 \in \overline{N_{\text{lot}}}] = \frac{\alpha(n-2)}{(n-1)} \frac{1}{2} = \frac{\alpha(n-2)}{2(n-1)}$$

If event $E_{\text{pred}}$ is not realized (and we are still conditioning on $\{1 \in \overline{N_{\text{lot}}}\}$), then this phase allocates to agent 1 with probability $\frac{1}{e}$ (as in mechanism $\mathcal{M}_{\text{phase}}$).

The corresponding probability is:

$$\Pr[t_{\text{pred}} \notin P \cap 1 \in \overline{N_{\text{lot}}}] = \Pr[t_{\text{pred}} \notin P \mid 1 \in \overline{N_{\text{lot}}}] \cdot \Pr[1 \in \overline{N_{\text{lot}}}] = \left( 1 - \frac{\alpha(n-2)}{(n-1)} \right) \frac{1}{2} = \frac{1}{2} - \frac{\alpha(n-2)}{2(n-1)}$$

Putting things together, we have argued that:

$$\Pr[E_{max}] = \Pr[E_{max} \mid t_{\text{pred}} \in P \cap 1 \in \overline{N_{\text{lot}}}] \cdot \Pr[t_{\text{pred}} \in P \cap 1 \in \overline{N_{\text{lot}}}]$$

$$+ \Pr[E_{max} \mid t_{\text{pred}} \notin P \cap 1 \in \overline{N_{\text{lot}}}] \cdot \Pr[t_{\text{pred}} \notin P \cap 1 \in \overline{N_{\text{lot}}}]$$

$$\geq \frac{1-2\alpha}{2e} + \frac{1}{2e(n-1)} \left( \alpha n - 1 + \frac{n}{2e} \right) \ln(1 + 2e\alpha).$$

□

For the lottery phase, as before, we lower bound the expected utility of this phase under event $E_{lot}$, which is the event that guarantees reaching the lottery phase without allocating, and with the balancedness property holding. Under mechanism $\mathcal{M}_{Pmax}$, $E_{lot} = E_{RO} \cap \bar{E}_{pred} \cap \{1 \in N_{lot}$ and $2 \in N_{sample}\}$, since event $\bar{E}_{pred} \cap \{1 \in N_{lot}$ and $2 \in N_{sample}\}$ precludes allocation in the max/prediction phase, and $E_{RO}$ guarantees balancedness. We highlight that, conditional on $\{1 \in N_{lot}$ and $2 \in N_{sample}\}$, the balanced property $E_{RO}$ is independent of $\bar{E}_{pred}$ (due to the symmetry of the balanced property). Note that the reason why we do not need to condition on agent 2 being the predicted highest value agent (as we did for mechanism $\mathcal{M}_{Psamp}$) is that this scenario can only improve the performance of the mechanism $\mathcal{M}_{Pmax}$ (since $\bar{E}_{pred}$ is guaranteed if we conditioned on $\{1 \in N_{lot}$ and $2 \in N_{sample}\}$). To compute the probability of $E_{lot}$, we first state a useful lemma.

**Lemma 11.** *Let $n \geq 27$ and let $\sigma$ be a uniformly random permutation of $[n]$. Let the prediction window be $P = [\frac{n}{2e} + 1 : (\frac{1}{2e} + \alpha)n]$ for some $\alpha \in (0, \frac{e-1}{2e})$. Assume the predicted highest agent is neither agent 1 nor agent 2. Then*

$$\Pr\big[\bar{E}_{pred} \cap \{1 \in N_{lot} \text{ and } 2 \in N_{sample}\}\big] = \frac{1}{4e} \cdot \frac{n}{n-1} \cdot \frac{n - \alpha n - 2}{n-2} \geq \frac{1-\alpha}{4e}$$

PROOF OF LEMMA 11.

We have already proven that

$$\Pr\big[1 \in N_{lot} \text{ and } 2 \in N_{sample}\big] = \frac{1}{4e} \cdot \frac{n}{n-1}$$

Out of the remaining $n - 2$ spots, $t_{pred}$ can be placed in $n - 2 - \alpha n$, thus:

$$\Pr[\bar{E}_{pred} \mid 1 \in N_{lot} \text{ and } 2 \in N_{sample}] = \frac{n - \alpha n - 2}{n-2}$$

As a result, we have argued:

$$\begin{aligned}
\Pr\big[\bar{E}_{pred} \cap \{1 \in N_{lot} \text{ and } 2 \in N_{sample}\}\big] &= \frac{1}{4e} \cdot \frac{n}{n-1} \cdot \frac{n - \alpha n - 2}{n-2} \\
&= \frac{1-\alpha}{4e} + \frac{(n-2) - \alpha(3n-2)}{4e\,(n-1)(n-2)} \\
&\geq \frac{1-\alpha}{4e},
\end{aligned}$$

where the last inequality is because $\alpha \leq \frac{e-1}{2e}$ and $n \geq 27$. □

Having computed this probability, we can now lower bound the probability of $E_{lot}$.

$$\begin{aligned}
\Pr[E_{lot}] &= \Pr[E_{RO} \mid \bar{E}_{pred} \cap \{1 \in N_{lot} \text{ and } 2 \in N_{sample}\}] \cdot \Pr[\bar{E}_{pred} \cap \{1 \in N_{lot} \text{ and } 2 \in N_{sample}\}] \\
&\geq \frac{4}{5} \frac{1-\alpha}{4e} = \frac{1-\alpha}{5e}
\end{aligned}$$

We can now compute the expected utility of $\mathcal{M}_{Pmax}$ similarly to before:

$$\begin{aligned}
\mathbb{E}[\text{Utility of } \mathcal{M}_{Pmax}] &\geq \mathbb{E}[\text{Utility of } \mathcal{M}_{Pmax}|E_{max}] \cdot \Pr[E_{max}] + \mathbb{E}[\text{Utility of } \mathcal{M}_{Pmax}|E_{lot}] \cdot \Pr[E_{lot}] \\
&\geq (v_1 - v_2) \cdot \left( \frac{1 - 2\alpha}{2e} + \frac{1}{2e(n-1)} \left( \alpha n - 1 + \frac{n}{2e} \right) \ln(1 + 2e\alpha) \right) \\
&\quad + \frac{1-\alpha}{5e} \frac{1}{25} \frac{\text{OPT}_L - (v_1 - v_2)}{5}.
\end{aligned} \tag{9}$$

We have thus expressed the robustness guarantee as a function of both $\alpha$ and $n$. Since the dependence on $n$ is monotone and only improves for larger $n$, it suffices to plug in any moderate lower bound on $n$ (for example, $n \geq 3$) in order to obtain a lower bound stated purely as a function of $\alpha$. □

**C.4. Comparison of $\mathcal{M}_{Psamp}$ and $\mathcal{M}_{Pmax}$.**

For a better comparison between the guarantees of the two bounds, we will take into account the contribution of terms $v_1 - v_2$ towards $\text{OPT}_L$. To this end, since $\text{OPT}_L \geq v_1 - v_2$, we introduce $\Delta = \frac{v_1 - v_2}{\text{OPT}_L}$, to rewrite our guarantees in a normalized form. Additionally, we will consider the comparisons asymptotically for $n$, to simplify a bit the expressions (most probability lower bounds that we computed thus far are essentially the expressions when $n$ goes to infinity). As we already discussed in the main body, we will have to make sure that the mechanisms are compared on the same consistency guarantee, and as such, we will be rewriting the guarantees of $\mathcal{M}_{Pmax}$ in terms of $\alpha = \frac{e^{2e\gamma}-1}{2e}$. Finally, to ease tedious calculations, we used symbolic algebra tools (the SymPy Python library) for algebraic transformations.

We start by rewriting the guarantees of $\mathcal{M}_{Psamp}$:

$$\mathbb{E}[\text{Utility of } \mathcal{M}_{Psamp}] \geq \frac{2(1 - 2e\gamma) \cdot \text{OPT}_L \; + \; \big(623 - (625 - 4e)\gamma\big) \cdot (v_1 - v_2)}{1250e}$$
$$= \frac{\text{OPT}_L}{1250e}\Big((2 + 623\Delta) + \gamma\big(-625\Delta + 4e(\Delta - 1)\big)\Big).$$

For $\mathcal{M}_{Pmax}$, we first look into the asymptotic probabilities of the event in Lemma 10:

$$\Pr[E_{max}] \geq \frac{1}{2e}\Big(1 - 2\alpha + \Big(\alpha + \frac{1}{2e}\Big)\ln(1 + 2e\alpha)\Big).$$

We recompute the expected utility guarantee of $\mathcal{M}_{Pmax}$ after substituting $\alpha = \frac{e^{2e\gamma}-1}{2e}$ and $\Delta = \frac{v_1 - v_2}{\text{OPT}_L}$.

$$\mathbb{E}[\text{Utility of } \mathcal{M}_{Pmax}] \geq (v_1 - v_2) \cdot \frac{1}{2e}\Big(1 - 2\alpha + \Big(\alpha + \frac{1}{2e}\Big)\ln(1 + 2e\alpha)\Big)$$
$$+ \frac{1 - \alpha}{5e}\frac{\text{OPT}_L - (v_1 - v_2)}{5} \cdot \frac{1}{25}$$
$$= \text{OPT}_L\left[\Delta\Big(\frac{e + 1 - e^{2e\gamma}}{2e^2} + \frac{\gamma e^{2e\gamma}}{2e}\Big) + \frac{1 - \Delta}{1250e^2}\Big(2e + 1 - e^{2e\gamma}\Big)\right].$$

For the comparison, the valid domain is $\gamma \in [0, 1/(2e)]$. As a sanity check, when $\gamma = 0$ we also have $\alpha = 0$, so both mechanisms have no prediction phase and recover the guarantee of $\mathcal{M}_{\text{phase}}$. Additionally, we can inspect the two endpoints $\Delta = 0$ and $\Delta = 1$.

Suppose first that $\Delta = 0$. Then the guarantees become

$$\mathbb{E}[\text{Utility of } \mathcal{M}_{Psamp} \mid \Delta = 0] \geq \text{OPT}_L\frac{1}{1250e}\Big(2 - 4e\gamma\Big),$$
$$\mathbb{E}[\text{Utility of } \mathcal{M}_{Pmax} \mid \Delta = 0] \geq \text{OPT}_L\frac{1}{1250e^2}\Big(2e + 1 - e^{2e\gamma}\Big).$$

We note that for all $\gamma$ values in the domain $\mathcal{M}_{Pmax}$ guarantee is weakly better than $\mathcal{M}_{Psamp}$.

Suppose next that $\Delta = 1$. Then the guarantees become

$$\mathbb{E}[\text{Utility of } \mathcal{M}_{Psamp} \mid \Delta = 1] \geq \frac{\text{OPT}_L(1 - \gamma)}{2e},$$
$$\mathbb{E}[\text{Utility of } \mathcal{M}_{Pmax} \mid \Delta = 1] \geq \text{OPT}_L\left[\frac{e + 1 - e^{2e\gamma}}{2e^2} + \frac{\gamma e^{2e\gamma}}{2e}\right].$$

Again, we note that for all $\gamma$ values in the domain $\mathcal{M}_{Pmax}$ guarantee is weakly better than $\mathcal{M}_{Psamp}$.

Thus far, at natural endpoints, we have identified that $\mathcal{M}_{Pmax}$ has weakly better guarantees than $\mathcal{M}_{Psamp}$. We extend this with the following algebraic lemma, which argues that in fact, for all values in the domain of $\gamma$ and $\Delta$, mechanism $\mathcal{M}_{Pmax}$ has a weakly better approximation guarantee than mechanism $\mathcal{M}_{Psamp}$.

**Lemma 12.** *Let $\Delta \in [0, 1]$ and $\gamma \in [0, 1/(2e)]$, and define*

$$F(\gamma, \Delta) = \Delta \left( \frac{e + 1 - e^{2e\gamma}}{2e^2} + \frac{\gamma e^{2e\gamma}}{2e} \right) + \frac{1 - \Delta}{1250e^2} \left( 2e + 1 - e^{2e\gamma} \right),$$

*and*

$$G(\gamma, \Delta) = \frac{(2 + 623\Delta) + \gamma \left( -625\Delta + 4e(\Delta - 1) \right)}{1250e}.$$

*Then $F(\gamma, \Delta) \geq G(\gamma, \Delta)$.*

*Proof.* Let $x \stackrel{\text{def}}{=} 2e\gamma$. Since $\gamma \in [0, 1/(2e)]$, we have $x \in [0, 1]$. A direct algebraic simplification gives

$$\begin{aligned}
1250e^2 \left( F(\gamma, \Delta) - G(\gamma, \Delta) \right) = {} & (1 - \Delta) \left( 1 + 2ex - e^x \right) \\
& + \frac{625}{2} \Delta \left( xe^x + x - 2e^x + 2 \right).
\end{aligned} \tag{10}$$

We show that both terms on the right-hand side are nonnegative.

First, define $f(x) = 1 + 2ex - e^x$. Since $f(0) = 0$ and $f'(x) = 2e - e^x \geq e > 0$ for all $x \in [0, 1]$, we get that $f(x) \geq 0$ on $[0, 1]$.

Second, define $h(x) = xe^x + x - 2e^x + 2$. Notice that $h(0) = 0$, $h'(x) = 1 + (x - 1)e^x$ and $h''(x) = xe^x \geq 0$ for all $x \in [0, 1]$. Since $h'(0) = 0$, it follows that $h'(x) \geq 0$ on $[0, 1]$, and hence $h(x) \geq 0$ on $[0, 1]$.

Since $\Delta \in [0, 1]$, both coefficients $(1 - \Delta)$ and $\Delta$ are nonnegative. Thus the right-hand side of Equation (10) is nonnegative, and therefore

$$F(\gamma, \Delta) \geq G(\gamma, \Delta).$$

$\square$

As a final point, we highlight that this subsection compares the two mechanisms according to the lower-bound analyses we have proved. Since these analyses are not tight, there may still be regimes where the actual performance of $\mathcal{M}_{Psamp}$ is better than the actual performance of $\mathcal{M}_{Pmax}$.

