# OpenReview forum: "Knowing Who, Not How Much: Learning-Augmented Mechanisms for Consumer Utility Maximization"
_ICML.cc/2026/Conference — ICML 2026 regular_

### Official Review · Reviewer_bNG9 · 2026-03-10

**Soundness:** 2
**Presentation:** 3
**Significance:** 2
**Originality:** 3
**Overall Recommendation:** 3
**Confidence:** 3

**Summary:**

This paper studies consumer utility maximization (residual surplus) in a single-item, online, random-order auction with strategic agents. The paper first gives a deterministic truthful three-phase mechanism for the random order setting, derived from offline random-sampling ideas, and claims a constant-factor approximation to the implementable benchmark OPTL. It then turns to learning-augmented mechanism design and makes the paper’s central conceptual point: predicting the top value is not enough, but predicting the identity of the top agent is enough to obtain best-of-both-worlds guarantees. Concretely, the paper proves an impossibility result for exact knowledge of FB =  $v_1$, and then analyzes two learning-augmented mechanisms, MPsamp and MPmax, that achieve constant consistency to FB when predictions are correct and constant robustness to OPTL when predictions are arbitrary. The comparison argues that MPmax yields a better consistency/robustness tradeoff than MPsamp under the paper’s lower-bound analysis.

**Compliance With Llm Reviewing Policy:**

Affirmed.

**Final Justification:**

Technical concerns raised have been addressed (although it requires quite a bit of work to revise). Given the dense nature of the writing, I’m not convinced the authors will be able to improve it to the standard of an ICML paper (and ensure correctness/accuracy) in this cycle. Thus I maintain my score as mildly negative.

**Key Questions For Authors:**

- Both learning-augmented mechanisms seem to top out around $1/(2e)$-consistency. Is there a matching impossibility for deterministic truthful mechanisms under this prediction model?
- Can you give a tighter or more interpretable robustness constant? Even a simple corollary that states an explicit numeric lower bound would help readers understand the real strength of the result.
- Can Lemma 2 be presented more transparently? Since so much depends on it, I would strongly encourage a cleaner statement/proof,

**Limitations:**

yes

**Strengths And Weaknesses:**

Strengths
- The paper identifies a qualitatively different kind of prediction for this objective: for consumer utility maximization, knowing who the best agent is matters, whereas knowing only how much the top value is does not. This cleanly differentiates this setting from prior learning-augmented mechanism design work.
- The three-phase mechanism Mphase is a nice construction.
- Nontrivial technical ideas


Weaknesses
- The main results are qualitatively “constant,” but the constants appear extremely small. For example, from Equation (3), the baseline robustness guarantee is only on the order of
1/(625e) in the worst case, and the best consistency obtainable by the learning-augmented mechanisms appears capped at about 1/(2e)
- The positive result assumes an online predictor that emits a binary signal exactly when the predicted top agent arrives. This is weaker than revealing all values, but it is still a fairly specialized signal model.
- The proof of Lemma 2 is the main place where I was not fully convinced on a first read. I suspect the statement may be true, but the current proof is hard to verify.
- Some important guarantees are hidden rather than stated explicitly. For instance, Theorem 1 says “constant C-approx” but without stating the constant in the theorem statement. Given how small the constants are, I think the paper should be more explicit about this in the main text.
- The high-level story is good, but several proofs are dense, notation is occasionally hard to parse.

Comments/Typo
- Line 131: defined if -> if
- Line 168 (definition of $N_max$): I don’t think the $s$ is supposed to be there right?
- Line 335: robustnesss -> robustness
- Line 510: there has a been -> there has been

---

> ### Author Rebuttal · Authors · 2026-03-30
>
> > The constants appear extremely small.
>
> The robustness constant carries over from the randomized offline approximation of HR08 (which is ~1/1500); our focus was not on improving constants. We instead focus on characterizing qualitatively what is possible in terms of approximation (and subsequently robustness) for the online random-order model for the OPT_L benchmark and further break the log n barrier to first-best in terms of consistency.
>
> > An online predictor that signals when the top agent arrives is still a fairly specialized model.
>
> Our prediction model uses very minimal information granularity. For instance, if there is a rich ML model about agents' preferences for this item, our mechanism only uses predictions of the form: when the model has high confidence that some person has the highest-value, it predicts yes. Please see our detailed response to reviewer r7Sm about the prediction model.
>
> > Some important guarantees are hidden rather than stated explicitly, e.g. “C-approx.”
>
> Using “C-approx” was meant to improve readability and to highlight the relationship between the robustness guarantee and the baseline guarantee without predictions. We made a mistake and didn’t express robustness as a function of C, but have since fixed this. However, if the reviewers feel that using constants would read better, we’re happy to do so, and also point out what carries over from the analysis of RSOL from HR08.
>
> > Q1: Consistency lower-bounds
>
> There is no matching impossibility for deterministic truthful mechanisms, and in fact, we can get much better consistency—just at the cost of robustness; it’s a tradeoff. For example, we can consider a 4-phase mechanism, where the first $n/2$ agents are used purely for prediction, and the remaining $n/2$ are proportionally allocated according to $M_{phase}$. This mechanism achieves 1/2-consistency, and constant robustness follows via a modified analysis: assuming the balancedness property between the first and second halves of $σ$, one can apply essentially the same analysis to the $M_{phase}$ portion, incurring only an additional constant-factor loss since the remaining utility in the second half is a constant-fraction of $OPT_L$ (following from Lemmas 4 and 5). We personally didn’t feel that this additional analysis added value, and it is a bit complicated, but we’d be happy to add commentary about this if the reviewers find it interesting.
>
> > Q2: Tighter or more interpretable robustness constant
>
> This is a really good point, thank you for raising this. In Appendix C.4, where we compare the robustness, we start with a more interpretable constant that is in terms of the ratio between $v_1-v_2$ and OPT_L. We will make sure to clean that up, bring it into the main body, and make the connections to the baseline C explicit.
>
> > Q3: Cleaner Lemma 2 proof
>
> We completely agree that our submission did not contain enough intuition regarding Lemma 2, but we have since updated the presentation to contain a thorough discussion. Here is some of the intuition:
>
> Balancedness is a property of the partition: for every $i$, the top-$i$ agents should be split so that neither side contains too small or large a fraction ($<i/6$ or $>5i/6$). One way to reason about this is to track the absolute difference between the sizes of the two sets as each element $i$ is placed in the partition. This induces a stochastic process that can be viewed as a random walk on the non-negative integers (initialized at 0), where a +1 step corresponds to placing the next element on the currently larger set, and a −1 step corresponds to placing it on the smaller set. Under this view, balancedness corresponds to the event that this walk never crosses a (history-dependent, time-varying) barrier ($2i/3$).
>
> In the random-subset (RSOL) process, each step of the walk is an independent move of ±1 with probability $1/2$. In contrast, under random-order ($M_{phase}$), the transition probabilities are history-dependent: at any point, the next assignment is biased toward the smaller set. That is, if the two processes are at the same state after some number of steps, the random-order walk is more likely to move toward 0 than the imbalance barrier. This single-step comparison captures the key intuition, but is not sufficient on its own to compare the full processes because of the history-dependence. This is precisely why we introduce intermediate processes, allowing us to compare the two processes at one step at a time (while sequentially conditioning on history), ultimately showing that balancedness under random order is more likely than under random subsets.
>
> > “The proof of Lemma 2 is the main place where I was not fully convinced on a first read.”
>
> Hopefully the above intuition helps, but if you have any specific questions that we can clarify, or places you think we should expand more, we would be very happy to hear. We also clarify certain notational/definition mishaps in our proof in response to reviewer dPe5.

---

> > ### Author Rebuttal · Reviewer_bNG9 · 2026-04-01
> >
> > I have read the rebuttal and maintain my score.

---

> > > ### Author Response · Authors · 2026-04-04
> > >
> > > Thank you for the follow-up! We’re glad to have addressed your questions. Since the acknowledgment mentions your concerns are resolved, yet you have stated your recommendation is a weak reject, if there are remaining issues that are affecting your final recommendation, it would be helpful for us to know so that we can address it in the revision.

---

### Official Review · Reviewer_x8Eq · 2026-03-13

**Soundness:** 3
**Presentation:** 2
**Significance:** 3
**Originality:** 3
**Overall Recommendation:** 4
**Confidence:** 4

**Summary:**

This paper studies consumer utility maximization in the online random-order model. Each agent (consumer) has a private valuation for the item. The goal of the mechanism is to allocate a single item to one of the nnn agents, who arrive in random order, so as to maximize consumer utility, namely valuation minus payment, rather than the seller’s revenue. The mechanism is DSIC, as it is implemented as a repeated posted-price auction.

The main motivation of the paper is to understand what kind of prediction oracle is actually useful in this setting. The paper studies two benchmarks: the first-best (FB) benchmark and the random lottery benchmark $OPT_L$. Without prediction, the achievable approximation ratio to the first-best benchmark, that is, the highest value, is logarithmic in nnn, while an $O(1)$ competitive ratio with respect to $OPT_L$​ is achievable. The paper then shows that this barrier remains even with a highest-value prediction oracle, that is, an oracle predicting $\max_i v_i$​. In contrast, with a highest-value-agent oracle, that is, an oracle predicting $\arg\max_i v_i$​, one can achieve both an $O(1)$ competitive ratio with respect to the FB benchmark, which the paper calls consistency, and an $O(1)$ competitive ratio with respect to the random lottery benchmark, which the paper calls robustness.

**Compliance With Llm Reviewing Policy:**

Affirmed.

**Final Justification:**

Aside from writing quality, I believe the result represents a solid contribution, hence I maintain my original positive score.

**Key Questions For Authors:**

1. Could the authors clarify how equation (1) follows from Lemmas 1 and 2? At present, I do not see the full argument, and an intermediate step seems to be missing.

2. How does this framework generalize to the revenue-maximization setting? Is there a similar finding there regarding the usefulness of different prediction oracles? More specifically, do we have similar results in revenue maximization showing that an identity oracle is more powerful than a max-value oracle?

- If not, what is the main technical hurdle in extending the present ideas and techniques to the revenue-maximization setting?

- If such results already exist (so far I haven't seen), then it would be helpful for the authors to clarify what is technically new here relative to that literature.

**Typos**
Line 208: $\bar{n}_i = | \bar{N}\cap\{1,2+,...,i\}|$ (no $+$)

Line 1131: Creflem:mpred-robustness

**Limitations:**

This work focuses on consumer utility maximization rather than revenue maximization, which is the more common setting in the literature. It would be interesting future work to see whether the same finding also extends to revenue maximization.

**Strengths And Weaknesses:**

### **Strengths**

The study of which prediction oracle is actually useful is interesting and relevant to ML-enhanced mechanism design. The paper highlights that different prediction oracles can have fundamentally different power by proving different lower and upper bounds on achievable competitive ratios. In that sense, the paper provides evidence for which kinds of predictive information are better suited to online mechanism design.
To the best of my knowledge, I have not seen such a clear distinction between the power of different predictive oracles in prediction-enhanced online mechanism design. For that reason, I think the paper makes a meaningful conceptual contribution and is worthwhile as an initial paper in this direction.

### **Weaknesses**
While the separation between different prediction oracles is interesting, it is less clear how the two oracle types perform in average-case settings. The results are worst-case competitive-ratio guarantees, but in practice such guarantees do not always match empirical performance. A small numerical study over standard valuation distributions, such as Gaussian or other common synthetic distributions, could strengthen the paper and help show the practical meaning of the theoretical separation.
Another weakness is the writing and presentation. Some parts are hard to follow, and some notation is introduced too suddenly. For example, the notation $N=\sigma[1:n/2]$ appears without enough explanation, and a clearer notation (like $N_\text{mid}$) or a more explicit definition would make the paper easier to read. More generally, I think the paper would benefit from a clearer structure and clearer explanation of the main ideas and proof steps.
There are also places where the arguments need more explanation. For example, I do not immediately see how equation (1) follows from Lemmas 1 and 2. From those lemmas, one can bound $\Pr[E_{RO}\mid n_1=0,n_2=1]\geq 4/5$, but some intermediate step seems to be missing. Writing this part more explicitly would make the proof easier to understand.

Overall, I think the paper has a meaningful contribution, but there is still room for improvement in writing and presentation. Clearer exposition would make the technical ideas and results easier to appreciate.

---

> ### Author Rebuttal · Authors · 2026-03-30
>
> > Could the authors clarify how equation (1) follows from Lemmas 1 and 2? At present, I do not see the full argument, and an intermediate step seems to be missing.
>
> We agree that clarity can be improved in the technical parts of our paper, and we will take measures to address issues in the final version of the paper. We clarify that $E_{3,1}$ is the event that agent 1 is in the lottery phase which is of size $n/2$ and that agent 2 is in the sample phase which is of size $n/2e$. Thus,
>
> 1. $\Pr[ E_{3,1}] = 1/2 \cdot (n/2e)/(n-1) > 1/4e$ because the probability of $v_1$ being in the lottery phase is $1/2$, and conditioned on this, the probability of $v_2$ being in the max phase is $(n/2e)/(n-1) > 1/2e$, and
>
> 2. $E_{3,1}$ implies event $\{n_1 = 0, n_2 = 1\}$ (line 223).
>
> 3. $\Pr [E_{RO} | n_1 = 0, n_2 = 1] \geq 4/5$ follows from Lemmas 1 and 2.
>
> > How does this framework generalize to the revenue-maximization setting? Is there a similar finding there regarding the usefulness of different prediction oracles? More specifically, do we have similar results in revenue maximization showing that an identity oracle is more powerful than a max-value oracle? If not, what is the main technical hurdle in extending the present ideas and techniques to the revenue-maximization setting? If such results already exist (so far I haven't seen), then it would be helpful for the authors to clarify what is technically new here relative to that literature.
>
> This literature is very well studied and surveyed in the introduction (but we do not explicitly mention that the objective is revenue—we will make sure to explicitly state this) and we have a more detailed survey in Appendix A (lines 510-517) due to space constraints by ICML. What is interesting about our work is how it differs from revenue. In short, in revenue, knowing how much (the value $v_1$) is enough! Balkanski et al. ‘24 consider revenue maximization in the online random order model, when having access to a prediction about the highest value $v_1$. They show constant consistency to the highest value $v_1$ and constant robustness to the second highest value $v_2$.
>
> There are other works that study revenue maximization in offline settings with predictions about each agent’s value (the full instance): Xu and Lu ‘22, Caragiannis & Kalantzis ‘24, Lu et al. ‘24.
>
> > While the separation between different prediction oracles is interesting, it is less clear how the two oracle types perform in average-case settings. The results are worst-case competitive-ratio guarantees, but in practice such guarantees do not always match empirical performance. A small numerical study over standard valuation distributions, such as Gaussian or other common synthetic distributions, could strengthen the paper and help show the practical meaning of the theoretical separation.
>
> This is a great question, and you actually raise a great point that we would be happy to add the paper. Our positive result under the predicting “who” model is prior-free, which is stronger than an average-case analysis, meaning that the same guarantees also hold under a distributional assumption. In contrast, the impossibility under predicting “how much” is proven under distributional assumptions (exponential distributions), highlighting an even sharper distinction between the two prediction models. This means that we have theoretically proven separation between the two prediction models under average-case settings as well.
>
> We also wanted to justify studying the prior-free setting in the following way (also stated in response to reviewer dPe5). In many real settings, such as advertising auctions, the seller does not have information regarding the buyers’ prior distributions. The prior-free setting is a well-established model to study this setting. (And, as illustrated above, because prior-free is a much weaker assumption, any results for this setting are much stronger.) However, the seller can collect and learn on information to form predictions. This is why we chose the predictions model. In addition, to be able to quantify the value of predictions, we must compare them to not having information, as opposed to already having distributional information—predictions will give little help over already having full distributional information.
>
> We totally understand the desire to empirically validate the results through a numerical study. Unfortunately, simulation requires choosing a distribution to draw our data from, so results will be average-case and therefore incomparable with the prior-free analysis we care about.
>
> > Another weakness is the writing and presentation.
>
> You’re absolutely right about the rushed presentation for the deadline; we apologize and have since worked on cleaning it up.

---

> > ### Author Rebuttal · Reviewer_x8Eq · 2026-04-04
> >
> > Thank you for the response. I encourage the authors to improve the quality and clarity of the writing. Aside from that, I believe the result represents a solid contribution, and I will maintain my original positive score.

---

### Official Review · Reviewer_r7Sm · 2026-03-13

**Soundness:** 3
**Presentation:** 4
**Significance:** 3
**Originality:** 2
**Overall Recommendation:** 4
**Confidence:** 3

**Summary:**

This paper studies how to maximize consumer utility in an online setting where strategic buyers arrive in a random order. Maximizing utility is notoriously hard because standard mechanisms must charge payments to keep buyers honest (truthful), but taking away money directly destroys the buyer's utility.

To bypass this impossibility, the authors use learning-augmented mechanism design (incorporating external predictions). They make a very sharp observation: past works usually predict the value of the item, but knowing the value doesn't help here because the platform still has to charge high prices to verify who actually holds that value. Instead, the authors identify that predicting the identity of the highest-valued buyer is the only effective way to protect utility.

The paper solves the problem in two main steps:

1.	A baseline mechanism: They first build a deterministic, truthful online mechanism by cleverly adapting offline randomized techniques.

2． Adding the predictor: They augment this baseline with the "identity prediction".

The final result guarantees the best of both worlds: a constant consistency (it achieves a constant approximation to FB when the prediction is right) and a constant robustness (it matches the best possible baseline performance even when the prediction is entirely wrong).

**Compliance With Llm Reviewing Policy:**

Affirmed.

**Final Justification:**

I am positive on this paper. I will keep my score as it is.

**Key Questions For Authors:**

Could you provide a deeper discussion on the relative information strength between predicting "who" and predicting "how much"? Specifically, under the objective of maximizing consumer utility, is knowing the exact identity ("who") fundamentally a stronger piece of information than knowing the optimal value ("how much")?

**Strengths And Weaknesses:**

Strengths：
1.	Excellent Presentation: The paper is exceptionally well-written, clear, and easy to follow.

2.	Interesting Technical Approach: The technical sections are solid. The methodology of adapting a randomized offline algorithm into a deterministic online mechanism is particularly clever and interesting to read.

3.	Complete and Solid Results: The theoretical results are complete and clearly stated, successfully achieving constant-factor approximations for both consistency and robustness.

Weaknesses:
1.	Predicting the exact identity of the highest-valued agent seems like an overly strong assumption for utility maximization. Is it realistic to know who values the item most before they even arrive? The paper needs concrete, real-world examples to justify why this specific prediction is practical.

---

> ### Author Rebuttal · Authors · 2026-03-30
>
> > Predicting the exact identity of the highest-valued agent seems like an overly strong assumption for utility maximization. Is it realistic to know who values the item most before they even arrive? The paper needs concrete, real-world examples to justify why this specific prediction is practical.
>
> The algorithms with predictions framework doesn’t assume the accuracy of the prediction model. For example, in the real world, some predictions might be really good for certain models, but not do well in other scenarios. We are providing worst-case guarantees that do well in both scenarios. In particular, if the prediction is correct, we capitalize on that, and if the prediction is not correct, we don’t want to be led astray.
>
> We do not assume that we know who values the item most before the agents even arrive. Our predictor signals the agent who has the highest value *only when* that agent arrives. Intuitively, if the predictor sees an agent with the “right features,” it might predict that they are the top agent with high confidence. And in fact our results can handle some error of the form: if the predicted agent’s value is a $\delta$-fraction of the highest value, then the consistency loses an additional factor $\delta$.
>
> Concretely, in an ad auction, Google has rich data on user features (demographics, browsing history, location, past purchases) as well as advertiser performance (historical click-through rates, conversation rates, ROI) and thus can predict with high confidence if the given advertiser has the highest value for showing an impression to a specific user, though they may not always be right.
>
> > Could you provide a deeper discussion on the relative information strength between predicting "who" and predicting "how much"? Specifically, under the objective of maximizing consumer utility, is knowing the exact identity ("who") fundamentally a stronger piece of information than knowing the optimal value ("how much")?
>
> Great question! What our paper shows is that for the objective of consumer utility, yes, knowing the exact identity ("who") is fundamentally a stronger piece of information than knowing the optimal value ("how much").
>
> In Theorems 3 and 4, we provided classes of mechanisms that achieve “best-of-both-worlds guarantees,” that is, (1) a constant approximation to FB when the prediction (of identity) is correct, and (2) a constant approximation to the optimal lottery when the prediction is incorrect. Yet in Theorem 2 we prove that a mechanism that precisely knows the optimal value (“how much”) can still not approximate FB within a constant factor (the gap is order $\log n$). Since the prediction may also be incorrect, this implies that any mechanism with a prediction of the optimal value cannot provide a best-of-both-worlds guarantee. Hence, a prediction of the identity is a fundamentally stronger piece of information than knowing the optimal value (for the objective of consumer utility).
>
> Intuitively, the exact identity allows the mechanism to identify the highest-valued agent without requiring high prices to do so, which aligns with the objective of utility. Interestingly, for a different objective such as revenue, this implies the opposite: knowing the optimal value is a stronger piece of information than knowing the exact identity. In fact, Balkanski et al. [2024] show that knowing how much $v_1$ is gives constant consistency (to $v_1$) and constant robustness (to $v_2$) for revenue maximization in a random-order model. This is in stark contrast to our Theorem 2, highlighting a fundamental difference between revenue maximization and utility maximization.

---

> > ### Author Rebuttal · Reviewer_r7Sm · 2026-04-04
> >
> > Thank you for your detailed response. I appreciate the clarification and will keep my score as is.

---

### Official Review · Reviewer_dPe5 · 2026-03-13

**Soundness:** 3
**Presentation:** 2
**Significance:** 3
**Originality:** 3
**Overall Recommendation:** 3
**Confidence:** 3

**Summary:**

This paper investigates the consumer utility maximization problem within an online random-order model. The fundamental challenge of mechanism designing in this setting arises from the tension in designing payment rules: while payments are necessary to elicit agents’ private values, they simultaneously reduce the consumer utility that the mechanism designer aims to maximize. The authors first adapt the Random Sampling Optimal Lottery of (Hartline & Roughgarden 2008) into a deterministic online mechanism, M_{phase}, proving that it maintains a constant-factor approximation to optimal lottery benchmark OPT_L. Then, the paper rigorously demonstrates that knowing the first best (FB) value is insufficient in overcoming the logarithmic barrier of FB. Finally, the authors construct two mechanisms which can leverage the prediction of the identity of the highest-valued agent, showing that this kind of information is enough in overcoming the logarithmic barrier when prediction is correct, while be robust (constant-factor approximation to the OPT_L) when prediction is wrong.

**Compliance With Llm Reviewing Policy:**

Affirmed.

**Key Questions For Authors:**

1. (i)(ii) in Weakness (1)

2. Weakness (2)

3. You mentioned that the proof of Theorem 2 closely resembles the analysis in (Qiao & Valiant, 2023). Could you please specify which aspects of your proof are new or different?

4. Could you explain your work’s limitation and suggest promising directions for future research?

**Limitations:**

Yes.

**Strengths And Weaknesses:**

Strengths:

(1)The consumer utility maximization problem is both interesting and under-explored

(2) The majority of the proofs in the appendix are rigorous and correct. However, I have a few reservations regarding some of them (please see the weaknesses section). I particularly appreciate the proof of Theorem 2, which I find both interesting and clever.

(3) Although the conclusion of learning the identity of the highest-valued agent is straightforward, I believe that formally presenting this statement is important. It provides valuable insight for the industry, emphasizing the importance of learning identities rather than focusing solely on values.

Weakness：

(1) The writing is not clear, for example,(i) why did you choose to consider the random-order model instead of the iid model?? (ii)What is the formal definition of deterministic mechanism. In the Liu et al. (2025) work you referenced (line 84), they define deterministic mechanisms as those whose allocation x_i \in {0,1}, which M_{toss} also satisfies. And what is the benefit of being deterministic; (iii) When taking expectations, the authors do not clarify what is being taken expectation over.

(2) The proof of lemma 2 appears to have some problem. Line 694 says “For j>0, Pr(N_{i+1})> 1/2> Pr(N_{i+1}=0) ”, what is the role of the variable j in this context, and why should this inequality hold universally? Does Pr(N_{i+1}) refer to the probability of assigning agent i+1 to N as defined in line 600? If so, I don’t think this inequality always holds, as it is history-dependent. Moreover, even if this inequality is assumed to be correct, you also states that  “Pr[\bar(E)_{R^{(i)}}]|state[i+1]=k+1] is greater than Pr[\bar(E)_{R^{(i)}}]|state[i+1]=k-1]” (line 691), the “\geq” in line 685 should be “\leq”, as the greater part gets larger probability.

(3) Lack of conclusion and discussion of future work.

---

> ### Author Rebuttal · Authors · 2026-03-30
>
> > Q1: Writing and precision
>
> We completely acknowledge that the writing is not clear and apologize; the submission was rushed for the deadline. We’ve been working on drastically improving the writing and precision. Thank you for pointing out these specific places that need improvement!
>
> (i) Random-order prior-free justification: In many real settings, such as advertising auctions, (a) agents arrive online and (b) the seller does not have a prior distribution. The random-order model and the prior-free setting are well-established models to study these settings. Also, because prior-free is a much weaker assumption, any results for this setting are much stronger, and hold for any setting with more information. However, the seller can collect and learn on information to form predictions. This is why we chose the predictions model. In addition, to quantify the value of predictions, we must compare to not having information—predictions will give little help over already having full distributional information.
>
> (ii) Deterministic mechanism: We have the same definition: $x_i\in\lbrace0,1\rbrace$, where $x_i$ is the expected allocation (in expectation over all coin flips in the mechanism). The *expected* allocation in our M_{toss} is not restricted to {0,1}, and is therefore not deterministic. For example, suppose we have an instance 4, 11, 10, 15. M_toss allocates to the last agent with probability 1/2 at price 11, and allocates to the 3rd agent with probability 1/2 at price 4. The benefit of deterministic mechanisms is that participants find them far more palatable for use in practice than randomized mechanisms; this is well-documented (e.g., the Nobel-prize winning work of Tversky Kahneman ‘92, or Liu et al. ‘25).
>
> (iii) Expectations: We will make sure this is fixed, thank you!
>
> > Q2: Lemma 2 proof issue
>
> Thank you for raising good points about the proof of Lemma 2. We emphasize that the lemma is correct, but we absolutely should have been clearer, and we have updated our paper to fix this. The main point is that at line 660, we introduce state[j], which captures how close the larger side of the process is to violating the upper boundary of balancedness. With that in mind, let us summarize how to address the particular points you raise:
> - We made a mistake and did not update our definitions. We should have appropriately defined the transition probability of line 669 “for steps at time $j\le i$” as $\min(\Pr[N_{j+1}|H_{j}],1-\Pr[N_{j+1}|H_{j}])$ (instead of $\Pr[N_{j}|H_{j-1}]$).
> - Then in lines 680-688, we need to change $Pr[S_i=1]=1/2$ (and $\Pr[S_i=0]=1/2$), and also change $\Pr[N_i=1]$ to the appropriately defined state transition probability defined above $\min(\Pr[N_{j+1}|H_{j}],1-\Pr[N_{j+1}|H_{j}])$ and similarly $\Pr[N_i=0]$ should be $\max(\Pr[N_{j+1}|H_{j}],1-\Pr[N_{j+1}|H_{j}])$.
> - Thus line 694 needs to instead argue $$\max(\Pr[N_{j+1}|H_{j}],1-\Pr[N_{j+1}|H_{j}])\ge 1/2\ge \min(\Pr[N_{j+1}|H_{j}],1-\Pr[N_{j+1}|H_{j}])$$ (which trivially holds). There was both a typo ($i$ instead of $j$) and a notational mishap on this line.
> - To address your point, $Pr(N_{i+1})$ is indeed history dependent, and its definition is the one in line 600.
> - Finally, with the interpretation of state we already have in the paper (the same emphasized here), the inequality in line 691 is correctly written as “greater.”
>
> We again apologize for the confusion. With these fixes, the spirit and arguments of the proof are correct.
>
> > Q3: Compare Thm 2 analysis with Qiao Valiant ‘23
>
> The main difference is that the [QV ‘23] result only holds for thresholds: posted prices cannot do better than a $\log n$ approximation. We, however, show this for any randomized mechanism, use tools from Bayesian mechanism design to do so, and relate this to consistency guarantees in mechanisms with predictions. That is, we build upon their truncated exponential example, but then compare against a stronger benchmark. The idea of using exponential samples is not originally from [QV ‘23], but is a commonly used tool throughout mechanism design (e.g., it is also used in [HR ‘08]).
>
> > Q4: Limitations and future directions
>
> Limitations: While we establish best-of-both-worlds tradeoffs asymptotically, we do not establish lower bounds (although these are elusive in algorithms with predictions) or tightness of tradeoffs by optimizing constants. Future directions: In addition to proving lower bounds, it would be good to generalize the best-of-both-worlds result to be error tolerant: on our current mechanisms if the predictor chooses an agent who is “close” to the highest value, then our results carry over with that corresponding error, but one might consider defining different metrics to capture imperfect but not arbitrarily bad predictions. Another direction is to extend our results beyond single item (e.g., k-units/large capacity, or even multi-dimensional). We are happy to add a conclusion and include open problems; thank you for the suggestion.

---

> > ### Author Rebuttal · Reviewer_dPe5 · 2026-04-03
> >
> > The author has addressed some of my questions, but overall the paper feels rather rushed. It is in need of significant revision in terms of writing and precision. I have therefore decided to keep my original score.

---

> > > ### Author Response · Authors · 2026-04-05
> > >
> > > We would like to again thank the reviewer for their thorough and helpful review as well as follow-up. Respectfully, we would like to argue that as the paper is technically "rigorous and correct" as well as "interesting and under-explored" by the reviewer's own evaluation, rejecting it only to improve exposition (which can be done by the camera-ready) would exclude a contribution that can be readily polished without altering its substance.
> > >
> > > In particular, as far as we are aware, the only substantive concern was regarding Lemma 2, which we resolved in our rebuttal by identifying an error carrying through notation. However, the underlying argument is unchanged and valid, and no claims of incorrectness or invalid results remain. In addition, the issues raised in Weakness (1), rather than indicating imprecision or errors, appear to stem primarily from conventions in the algorithmic game theory literature, where certain common modeling choices, definitions, and expectations are implicit for expert readers. That said, we agree that making these details explicit will improve accessibility, and we appreciate the reviewer's suggestions for clarifying these points.
> > >
> > > We agree with the reviewer that the paper can be improved (and have already begun substantial revisions, see, e.g., our response to Reviewer bNG9's Q3). However, we believe that the presentation of our submission is sufficiently clear to fully evaluate our work: the reviewers were clearly able to engage deeply with the technical content, ask detailed questions, assess correctness and significance, and provide a comprehensive review which indicates that the paper is readable and its contributions are accessible, suggesting the writing as fixable exposition rather than a barrier to understanding. Writing quality, while important, is fully correctable in the camera-ready version and does not affect the validity of the results (prioritized by ICML for correctness, novelty, and significance, and evaluated "good" by the reviewer in each of these dimensions).

---

### Decision · Program_Chairs · 2026-04-30

**Decision:**

Accept (regular)

**Comment:**

The paper studies the problem of residual surplus maximization (i.e., agents' utility maximization, rather than welfare maximization) in a single-item setting where the agents arrive in a random order. The paper seeks to circumvent strong impossibility results in this setting by having ML predictions. It shows that having a prediction of the highest value is not of much help. Rather, having a prediction of the agent with the highest value helps. This is a cool insight. The paper develops mechanisms to establish the consistency-robustness results, namely, that their mechanism does well both when the predictions are right and when they are arbitrarily bad.

The review team liked the problem studied and the technical contribution of the paper. There were some concerns about writing, particularly a proof, that was clarified by the authors in their rebuttal. There were also concerns about the nature of the prediction itself: i.e., being able to meaningfully predict the identity of the highest-valued agent seemed like a tall ask (yes, we understand that the predictions need not be accurate, but if they are totally useless, there is no point in having them at all). But here, I am not too concerned: with reasonable feature access, one can aim to have a good prediction of whether the current agent is the highest valued agent, even without seeing other agents.

Overall, this is a decent paper to appear in ICML.